# ConMix: Contrastive Mixup at Representation Level for Long-tailed Deep Clustering

**Zhixin Li[1], Yuheng Jia[1,2]\***

[1] School of Computer Science and Engineering, Southeast University, Nanjing 210096, China
[2] Key Laboratory of New Generation Artificial Intelligence Technology and Its
Interdisciplinary Applications (Southeast University), Ministry of Education, China
`{lizhixin,yhjia}@seu.edu.cn`

## Abstract

Deep clustering has made remarkable progress in recent years. However, most existing deep clustering methods assume that distributions of different clusters are balanced or roughly balanced, which are not consistent with the common long-tailed distributions in reality. In nature, the datasets often follow long-tailed distributions, leading to biased models being trained with significant performance drop. Despite the widespread proposal of many long-tailed learning approaches with supervision information, research on long-tailed deep clustering remains almost uncharted. Unaware of the data distribution and sample labels, long-tailed deep clustering is highly challenging. To tackle this problem, we propose a novel contrastive mixup method for long-tailed deep clustering, named ConMix. The proposed method makes innovations to mixup representations in contrastive learning to enhance deep clustering in long-tailed scenarios. Neural networks trained with ConMix can learn more discriminative representations, thus achieve better long-tailed deep clustering performance. We theoretically prove that ConMix works through re-balancing loss for classes with different long-tailed degree. We evaluate our method on widely used benchmark datasets with different imbalance ratios, suggesting it outperforms many state-of-the-art deep clustering approaches. The code is available at `https://github.com/LZX-001/ConMix`.

## 1 Introduction

Despite the rapid advancements in deep clustering in recent years, these methods often struggle to be directly applied in real-world scenarios due to a lack of consideration for the long-tailed distributions. Unlike the commonly used uniform distributed datasets in labs, the datasets in nature usually present Zipf long-tailed distributions over classes, where major classes (head classes) have more samples compared to minor classes (tail classes) (Feldman, 2020). Training with class-imbalanced datasets naturally leads to biased models with significant performance drop (Hou et al., 2023). Methods with supervision information have been continuously proposed to eliminate the negative impact of long-tailed distributions on model training. Re-sampling (Ando and Huang, 2017; Han et al., 2005; He and Garcia, 2009; Shi et al., 2023) balances the number of samples for each class participating in model training. Re-weighting (Cui et al., 2019; Park et al., 2021; Lin et al., 2017; Ren et al., 2020; Tan et al., 2020) aims to re-balance loss of different classes through different weights. Logit adjustment (Menon et al., 2020; Tian et al., 2020; Wu et al., 2021) seeks to adjust prediction logits to correct biased models. Although these methods and their sequels can alleviate the performance drop caused by long-tailed distributions, they are all label-dependent and unable to directly use for unsupervised learning. As without the supervision information, it will be difficult to accurately distinguish between head class and tail class samples. Inspired by (Hooker et al., 2019), SDCLR (Jiang et al., 2021) prunes network to identify difficult-to-memorize samples, usually atypical and rare samples. In long-tailed self-supervised learning, SDCLR achieves good performance, but atypical and rare samples cannot be equated with tail class samples. Due to the large number of samples, there are also many atypical samples in the head classes, and tail class samples may

---

*Corresponding author: Yuheng Jia.

not necessarily be rare. Meanwhile, the common pseudo labeling in deep clustering is not accurate under long-tailed distributions, as distance-based labeling methods like K-means (Hartigan and Wong, 1979) generally lead to uniform distribution results. Therefore, difficulties in differentiating head class and tail class samples pose challenges in extending label-dependent long-tailed learning to long-tailed deep clustering.

In recent years, deep clustering has been witnessed sustained development due to the advancement of self-supervised representation learning (Van Gansbeke et al., 2020; Niu et al., 2022; Huang et al., 2023; Li et al., 2021b; 2022; Yu et al., 2023; Shen et al., 2021). But unlike self-supervised learning that can be further fine-tuned with labels, deep clustering lacks supervision information, thus is more severely affected by long-tailed distributions. Given long-tailed deep clustering is naturally more challenging than deep clustering on uniform datasets, few methods propose solutions on long-tailed deep clustering. We notice a very recent work $P^2OT$ (Zhang et al., 2024) makes contributions to express pseudo labeling by progressive partial optimal transport for imbalance clustering. However, the method is based on pre-trained DINO (Caron et al., 2021) and is to fine-tune the last block of ViT-B16 (Dosovitskiy et al., 2020) instead of training from scratch. Hence, the bias affected by the long-tailed distribution in the overall model is relatively weak. In contrast, the purpose of our work is to propose a method that can train a model from scratch while alleviating the affect of the long-tailed distributions.

The challenges in long-tailed deep clustering is primarily on three main points. (i) The lack of label information renders us unable to directly differentiate between samples from head classes and tail classes, as well as leading to lack evidence that are conducive to distinguishing samples of different classes. (ii) While experimental observations can identify difficult-to-memorize (e.g., SDCLR (Jiang et al., 2021)) instances, these do not necessarily equate to tail class samples. (iii) Prior research (Zhang et al., 2023) has highlighted the feature space occupied by samples from head classes is often larger than that of tail classes, whereas existing clustering algorithms tend to produce clusters that are balanced in sizes. Under such extremely constrained conditions, we only know that tail class samples are scarce compared to the abundance of head class samples. The core of the problem becomes whether we can implicitly leverage the inherent characteristics of long-tailed distributions to enhance deep clustering.

Mixup (Zhang et al., 2018) always leads to robustness and generalization in supervised learning (Zhang et al., 2021). In long-tailed supervised learning, mixup has been proved to benefit representation learning by improving complexity and diversity of datasets (Zhong et al., 2021). However, in a unsupervised manner, input-level mixup may harm representation learning due to the inability to learn semantically meaningful representations for clustering. Therefore, Manifold Mixup (Verma et al., 2019), a supervised method which interpolates representations in the feature space for better representations, is referable.

Inspired by Manifold Mixup (Verma et al., 2019), we propose to extend multi-sample mixup at representation level to long-tailed deep clustering. In detail, we multi-sample in every batch and interpolate representations learned in contrastive learning to synthesize new representations. Unlike previous works (Kalantidis et al., 2020; Zhang et al., 2022) only using the synthesized representations to form negative pairs with the original representations, we directly use synthesized representations for further learning. Neural networks learned in this manner can obtain discriminative representations to benefit long-tailed deep clustering. We name the proposed method ConMix, a shorthand for **Con**trastive **Mix**ing.

In summary, our main contributions is as follows:

• We propose a contrastive mixup at representation level for long-tailed deep clustering. In this manner, models can effectively alleviate the bias caused by long-tailed distributions and obtain competitive clustering performance.

• Most existing mixup methods primarily focus on supervised learning and input-level mixing. We have found an effective multi-sample representation-level mixup method that can be used for directly training networks in unsupervised learning thus extend it to deep clustering.

• We provide theoretical proof of the proposed method. In unsupervised conditions, it is very difficult to distinguish between head class and tail class samples and find a right antidote. But our method

can implicitly re-balance the losses of head and tail classes, which is similar to many long-tailed supervised learning approaches.

## 2 RELATED WORK

### 2.1 DATA IMBALANCE

In reality, data distributions always follow long-tailed distributions where head classes are dominate and tail classes are minor. In long-tailed distributions, supervised learning methods tend to overfit samples from head classes and underfit those from tail classes, leading to poor generalization performance. To address this issue, numerous methods have been proposed, including re-sampling (Ando and Huang, 2017; Han et al., 2005; He and Garcia, 2009; Shi et al., 2023), re-weighting (Cui et al., 2019; Park et al., 2021; Lin et al., 2017; Ren et al., 2020; Tan et al., 2020), and logit adjustment (Menon et al., 2020; Tian et al., 2020; Wu et al., 2021; Jia et al., 2024). Besides, methods (Kang et al., 2020; Zhong et al., 2021; Hou et al., 2023; Chu et al., 2020) based on two-stage decoupled training scheme, which decouple representation learning and classifier training, have achieved excellent performance in recent years. However, these methods rely on the prior provided by labels in long-tailed datasets, which is absent in unsupervised scenarios. Very few self-supervised methods, including (Jiang et al., 2021; Zhou et al., 2022; Liu et al., 2021), propose solutions for addressing class imbalance specifically. Usually, they implicitly increase the weights for rare samples training. These prove effective under self-supervised frameworks but do not readily translate to long-tailed deep clustering, where the aim is to enhance discriminability for all samples, spreading out the distribution of different classes while those from the same class to cluster closely together, rather than merely improving the learning of underrepresented ones. As for long-tailed clustering, the first work is (Zhang et al., 2024), which trains and fine-tunes on pre-trained DINO (Caron et al., 2021). However, to our knowledge, no method trained on long-tailed datasets from scratch has been proposed. Our objective is to delve deeper into the implications of long-tailed distributions on deep clustering and figure out how to solve it. Therefore, it is necessary to train models from scratch on long-tailed datasets that are specifically designed for this task.

### 2.2 DEEP CLUSTERING

Deep clustering aims to learn good representations and cluster samples in an unsupervised manner. DEC (Xie et al., 2016) and its variants (Li et al., 2018; Yang et al., 2017; Peng et al., 2023; 2021) train the model iteratively by autoencoder reconstruction loss. Although these methods can achieve better performance than traditional clustering, deep clustering methods based on autoencoders find it difficult to capture the complex structural features of input samples. Some methods based on similarity between samples (Chang et al., 2017; Ji et al., 2019; Tao et al., 2020) are proposed for more accurate partitions. Pseudo-labeling-based methods (Van Gansbeke et al., 2020; Tian et al., 2017; Niu et al., 2022; Xu et al., 2023) employ pseudo labels in deep clustering to guide network training. But how to obtain pseudo labels of both high quality and quantity is a problem that needs to be considered. Due to the advancement of contrastive learning framework, including SimCLR (Chen et al., 2020), Moco (He et al., 2020), BYOL (Grill et al., 2020), etc, contrastive-based deep clustering has been massive proposed, including (Van Gansbeke et al., 2020; Niu et al., 2022; Huang et al., 2023; Li et al., 2021b; 2022; Yu et al., 2023; Shen et al., 2021). They either simply use contrastive learning as representation learning, then train pseudo-label-based classifiers on good features, or propose contrastive-learning-based algorithms to make representations more discriminative, or directly output clustering results based on contrastive learning. For example, SCAN (Van Gansbeke et al., 2020) first trains a feature extraction network by SimCLR for subsequent pseudo-labeling; ProPos (Huang et al., 2023) innovatively integrates prototype-level SimCLR and instance-level BYOL; CC (Li et al., 2021b) is based on SimCLR and directly train a prediction network for clustering.

### 2.3 MIXUP

Mixup (Zhang et al., 2018) has demonstrated its efficacy in improving model generalization in supervised learning. Generally, Mixup regularizes the model by interpolations of the inputs and corresponding labels. Its variants (Uddin et al., 2021; Yun et al., 2019; Yao et al., 2022) make advancements on the form of interpolation or the strategy for sample selection. On the other hand, Mani-

fold Mixup (Verma et al., 2019) interpolates in the feature space to guarantee a smoother decision boundary. Interpolations in the feature space inspire some mixup-based methods in self-supervised learning. In some methods (Kalantidis et al., 2020; Zhang et al., 2022), feature space mixups are used to generate hard negative samples for contrastive learning.

# 3 METHOD

## 3.1 PRELIMINARIES

**Contrastive Learning.** Contrastive Learning, including SimCLR (Chen et al., 2020) and SDCLR (Jiang et al., 2021) learns instance-level visual representations via pulling positive pairs close and pushing negative pairs away. We assume an instance-level representation $v_i$. $v_i^+$ forms its positive pair and $v_i^- \in \mathbb{V}_i^-$ are included in negative pairs, where $\mathbb{V}_i^-$ is a set of negative samples for $v_i$. Then contrastive loss in SimCLR can be defined as:

$$\mathcal{L}_{\text{CL}} = \frac{1}{N} \sum_{i=1}^{N} -\log \frac{s\left(v_i, v_i^+, \tau\right)}{s\left(v_i, v_i^+, \tau\right) + \sum_{v_i^- \in \mathbb{V}_i^-} s\left(v_i, v_i^-, \tau\right)}, \tag{1}$$

where $\tau$ is a temperature parameter and $s(u, v, \tau) = \exp(u^\top v / \tau)$ when $u$ and $v$ are $\ell_2$-normalized representations. Usually, $v_i$ and $v_i^+$ are different augmented versions of the same input sample and $v_i^-$ are other representations in the batch, excluding $v_i$ and $v_i^+$.

**Manifold Mixup**. Manifold Mixup (Verma et al., 2019) linearly interpolates feature in hidden layers to encourage flatter class representations that possess better generalization. As a supervised method, the mixup of feature $z_i$ and $z_j$ is defined as:

$$\begin{aligned} \widetilde{z}_{syn} &= \lambda \cdot z_i + (1 - \lambda) \cdot z_j \\ y_{\widetilde{z}_{syn}} &= \lambda \cdot y_{z_i} + (1 - \lambda) \cdot y_{z_j}, \end{aligned} \tag{2}$$

where $y_a$ represents the label vector of $a$ and mixing coefficient $\lambda \sim \text{Beta}(\alpha, \alpha)$.

We extend Manifold Mixup to contrastive-learning-based deep clustering to reduce model bias caused by long-tailed distributions. Considering that mixup at the input level may mislead unsupervised models, we instead sample semantic representations to synthesize new mixed ones. Meanwhile, to obtain more diverse synthetic features, we choose multi-sampling instead of pairwise sampling.

## 3.2 MOTIVATION

In recent years, deep long-tailed learning has made great progress, for the purpose that deep learning models can be better suited to real-world application scenarios (Zhang et al., 2023). However, the majority of proposed methods focus on learning with supervision information. Without label information, it becomes hugely challenging to perceive the long-tailed distributions. Hence, very few works are dedicated to long-tailed unsupervised learning, especially long-tailed deep clustering, as introduced in Section 1 and 2. As long-tailed self-supervised methods focus on training hard-to-memorize samples, a method for deep clustering should aim at enhancing the handling of overall long-tailed distributions therefore mitigating model bias. Inspired by Manifold Mixup (Verma et al., 2019) in Eq. (2), as well as previous practice of mixup in long-tailed supervised learning (Zhong et al., 2021), we propose a mixup-based method under the framework of contrastive learning, which significantly enhances the model training on long-tailed distributions, even outperforms state-of-the-art deep clustering methods.

## 3.3 CONMIX

In our method, we follow the framework of SimCLR (Chen et al., 2020), including its network structure, data augmentation, and loss function. Consider a unlabeled long-tailed dataset of size $N$: $\{x_1, x_2, ..., x_N\}$. In SimCLR framework, each input $x_i$ is data-augmented twice, and the two augmented versions are fed into two different network branches in SimCLR. Given $N$ inputs, the network will output $2N$ representations $\{v_1, v_2, ...v_{2N}\}$. Let us assume, for each $i \in [1, N]$, $v_i$

and $\boldsymbol{v}_{i+N}$ are positive pairs from the same input while others are negative pairs. In representations from distinct inputs $\{\boldsymbol{v}_1, \boldsymbol{v}_2, ...\boldsymbol{v}_N\}$, we randomly generate $M$ synthesized representations $\{\boldsymbol{z}_1, \boldsymbol{z}_2, \boldsymbol{z}_3, ...\boldsymbol{z}_M\}$ in below manner:

$$\boldsymbol{z}_m = \frac{\bar{\boldsymbol{z}}_m}{\|\bar{\boldsymbol{z}}_m\|_2}, \quad \text{where} \quad \bar{\boldsymbol{z}}_m = \frac{1}{|\mathbb{U}_m|} \sum_{k \in \mathbb{U}_m} \boldsymbol{v}_k, \tag{3}$$

where $\|\cdot\|_2$ is the $\ell_2$-norm, enabling the synthesized representations available for contrastive learning, $\mathbb{U}_m$ is a set of indexes belonged to original representations which synthesize $\boldsymbol{z}_m$ and $|\cdot|$ denotes the number of elements in the set. In each batch, we randomly assign tags within $[1, M]$ to original representations from the same network branch $\{\boldsymbol{v}_1, \boldsymbol{v}_2, ...\boldsymbol{v}_N\}$. The generation of tags follows a uniform distribution with equal probabilities $\frac{1}{M}$ and original representations with the same tag are used to synthesize one particular representation in the manner of Eq. (3). This multi-sampling schedule results in synthesized representations with different numbers of $\boldsymbol{v}_i$. Not only does it obtain more diverse synthesized representations, but it also assigns different weights to different samples, similar to how mixup does. We use the exact same method to synthesize $\{\boldsymbol{z}_{1+M}, \boldsymbol{z}_{2+M}, \boldsymbol{z}_{3+M}, ...\boldsymbol{z}_{2M}\}$ from $\{\boldsymbol{v}_{1+N}, \boldsymbol{v}_{2+N}, \boldsymbol{v}_{3+N}, ...\boldsymbol{v}_{2N}\}$, including the same random tags, ensuring that $\boldsymbol{z}_i$ and $\boldsymbol{z}_{i+M}$ are generated from the same samples of different augmentations. In above manner, we extend the original positive and negative sample pairs to the synthesized positive and negative sample pairs: $\boldsymbol{z}_i$ and $\boldsymbol{z}_{i+M}$ are the new positive pairs synthesized from the same input samples while others are negative. For simplicity, $\boldsymbol{z}_i^+$ represents the positive sample of $\boldsymbol{z}_i$. Then the contrastive loss in ConMix can be formulated as below:

$$\mathcal{L}_{\text{CM}} = \frac{1}{2M} \sum_{i=1}^{2M} -\log \frac{s\left(\boldsymbol{z}_i, \boldsymbol{z}_i^+, \tau\right)}{\sum_{k=1}^{2M} \mathbf{1}_{[k \neq i]} s\left(\boldsymbol{z}_i, \boldsymbol{z}_k, \tau\right)}, \tag{4}$$

where $\mathbf{1}_{[k \neq i]}$ is an indicator function. It equals 1 when $k \neq i$, otherwise 0. In the ConMix framework, the loss contrast average features of multiple representations instead of individual representations. It can be seen as a augmented form of mixup at representation level.

# 4 THEORY

***ConMix can improve the performance of long-tailed deep clustering without the need to distinguish between head and tail samples in advance because it can re-balance the loss between head and tail classes.*** In the existing work on long-tailed supervised learning (Hou et al., 2023; Kang et al., 2020; Menon et al., 2020), many approaches focus on balancing the loss across different classes, that is, artificially reducing the loss for head class samples and increasing the loss for tail class samples, in order to correct for biased models. These methods are relatively simple to implement when supervised information is available, but it becomes much more challenging in an unsupervised setting. As ConMix mixups representations from different classes, it can implicitly achieve the loss-balance via improving the loss of tail class samples and reducing the loss of head class samples. We have uncovered this perspective through theoretical analysis. The following is our proof.

We first rewrite the $\mathcal{L}_{\text{CL}}$ in a form alike $\mathcal{L}_{\text{ConMix}}$ and then simplify it (the detailed derivation is in Appendix C.1) :

$$\mathcal{L}_{\text{CL}} = \frac{1}{2N} \sum_{i=1}^{2N} \log \sum_{k=1}^{2N} \mathbf{1}_{[k \neq i]} \exp(f_{\text{CL}}(\boldsymbol{v}_i, \boldsymbol{v}_k, \boldsymbol{v}_i^+)/\tau), \tag{5}$$
$$\text{where } f_{\text{CL}}(\boldsymbol{v}_i, \boldsymbol{v}_k, \boldsymbol{v}_i^+) = \boldsymbol{v}_i^\top (\boldsymbol{v}_k - \boldsymbol{v}_i^+).$$

In Eq. (5), $f_{\text{CL}}(\boldsymbol{v}_i, \boldsymbol{v}_k, \boldsymbol{v}_i^+)$ calculates the similarity difference between each sample $\boldsymbol{v}_i$ with its particular negative sample $\boldsymbol{v}_k$ and with positive sample $\boldsymbol{v}_i^+$. For a specific $\boldsymbol{v}_i$, its loss value depends on the difference between the similarity of all negative sample pairs and the similarity of positive pair. However, since $f_{\text{CL}}(\boldsymbol{v}_i, \boldsymbol{v}_k, \boldsymbol{v}_i^+)$ cannot measure the contribution of individual representation to the total loss, so we propose a new function $h_{\text{CL}}(\boldsymbol{v}_i)$:

$$h_{\text{CL}}(\boldsymbol{v}_i) = \sum_{k=1}^{2N} \mathbf{1}_{[k \neq i]} f_{\text{CL}}(\boldsymbol{v}_i, \boldsymbol{v}_k, \boldsymbol{v}_i^+) = \sum_{k=1}^{2N} \mathbf{1}_{[k \neq i]} \boldsymbol{v}_i^\top (\boldsymbol{v}_k - \boldsymbol{v}_i^+). \tag{6}$$

In the Eq. (6), the larger $h_{\mathrm{CL}}(\boldsymbol{v}_i)$ is, the larger the sum of $f_{\mathrm{CL}}(\boldsymbol{v}_i,,\boldsymbol{v}_i^+)$s associated with the specific $\boldsymbol{v}_i$ is, thus the greater the loss associated with $\boldsymbol{v}_i$ is in $\mathcal{L}_{\mathrm{CL}}$. In other word, $h_{\mathrm{CL}}(\boldsymbol{v}_i)$ indicates the contribution of representation $\boldsymbol{v}_i$ to the total $\mathcal{L}_{\mathrm{CL}}$. From the Eq. (6), we can see that in a long-tailed distribution, head class representations always have greater loss for more similar negative pairs from the same class, leading to a biased model. On the contrary, samples from tail classes, due to their scarcity of similar instances, tend to incur smaller loss and are consequently overlooked during training.

As for ConMix, we also need to measure the impact of individual $\boldsymbol{v}_i$, but the situation is slightly different. $\mathcal{L}_{\mathrm{CM}}$ can be similarly rewritten as a form of $f_{\mathrm{CM}}(\boldsymbol{z}_i, \boldsymbol{z}_k, \boldsymbol{z}_i^+)$:

$$
\mathcal{L}_{\mathrm{CM}} = \frac{1}{2M} \sum_{i=1}^{2M} \log \sum_{k=1}^{2M} \mathbf{1}_{[k \neq i]} \exp(f_{\mathrm{CM}}(\boldsymbol{z}_i, \boldsymbol{z}_k, \boldsymbol{z}_i^+)/\tau),
$$

$$
\text{where } f_{\mathrm{CM}}(\boldsymbol{z}_i, \boldsymbol{z}_k, \boldsymbol{z}_i^+) = \sum_{k=1}^{2N} \mathbf{1}_{[k \neq i]} \boldsymbol{z}_i^\top (\boldsymbol{z}_k - \boldsymbol{z}_i^+),
$$

(7)

To measure the impact of an individual $\boldsymbol{v}_i$ on $\mathcal{L}_{\mathrm{CM}}$, we assume that $\boldsymbol{z}_x$ is the synthesized representation corresponding to $\boldsymbol{v}_i$, and calculate the difference in all $f_{\mathrm{CM}}(\boldsymbol{z}_x,, \boldsymbol{z}_x^+)$s when $\boldsymbol{v}_i$ synthesizes $\boldsymbol{z}_x$ versus when it does not (the detailed derivation is described in Appendix C.2):

$$
h_{\mathrm{CM}}(\boldsymbol{v}_i) = \sum_{y=1}^{2M} f_{\mathrm{CM}}(\boldsymbol{z}_x, \boldsymbol{z}_y, \boldsymbol{z}_x^+) - \sum_{y=1}^{2M} f_{\mathrm{CM}}(\boldsymbol{z}_x', \boldsymbol{z}_y, \boldsymbol{z}_x'^+)
$$

$$
:= \sum_{k=1}^{2N} \boldsymbol{v}_i^\top \boldsymbol{v}_k - \sum_{k \in \mathbb{S}_i} \boldsymbol{v}_i^\top \boldsymbol{v}_k - (2M+1) \sum_{k \in \mathbb{S}_i^+} \boldsymbol{v}_i^\top \boldsymbol{v}_k,
$$

(8)

where $\boldsymbol{z}_x'$ is synthesized from all representations composed of $\boldsymbol{z}_x$ except $\boldsymbol{v}_i$. $\mathbb{S}_i$ and $\mathbb{S}_i^+$ are the sets contain indexes of original representations from $\boldsymbol{z}_x$ and $\boldsymbol{z}_x^+$ respectively, as $\boldsymbol{v}_i$ synthesizes $\boldsymbol{z}_x$ and $\boldsymbol{v}_i^+$ synthesizes $\boldsymbol{z}_x^+$. := denotes the assignment operator. Note that we have made simplification to omit the normalization of the synthesized representations and the proposed $h_{\mathrm{CM}}$ is adapted to align with $h_{\mathrm{CL}}$ for better comparison. $h_{\mathrm{CM}}(\boldsymbol{v}_i)$ indicates the contribution of $\boldsymbol{v}_i$ to $\mathcal{L}_{\mathrm{CM}}$, and larger $h_{\mathrm{CM}}(\boldsymbol{v}_i)$ leads to larger $\mathcal{L}_{\mathrm{CM}}$. Based on the distinction in similarity (inner product) of representations across different classes, now we can propose a new theorem to explain how ConMix re-balances the loss between head and tail class samples.

**Theorem 1.** *Let the similarity between representations within the same class is roughly the same $s_w$ and the similarity between samples of different class is also roughly the same $s_b$. $s_w > s_b$. And in ConMix, the number of original representations composing each synthesized representation is roughly equal, and their class distribution aligns with the overall class distribution. Then in the long-tailed distributions, for a head class representation $\boldsymbol{v}_h$ and a tail class representation $\boldsymbol{v}_t$, $h_{\mathrm{CM}}(\boldsymbol{v}_h) - h_{\mathrm{CM}}(\boldsymbol{v}_t) < h_{\mathrm{CL}}(\boldsymbol{v}_h) - h_{\mathrm{CL}}(\boldsymbol{v}_t)$.*

The detailed proof of **Theorem 1** is in the Appendix C.3. Given that $h_{\mathrm{CM}}(\boldsymbol{v}_h) - h_{\mathrm{CM}}(\boldsymbol{v}_t) < h_{\mathrm{CL}}(\boldsymbol{v}_h) - h_{\mathrm{CL}}(\boldsymbol{v}_t)$, the loss of $\boldsymbol{v}_h$ is relatively lower in ConMix compared to contrastive learning, and $\boldsymbol{v}_t$ has larger loss in ConMix relatively compared to contrastive learning. ***That is, ConMix can relatively lower the loss of head class samples and increase the loss of tail class samples, thus achieve loss re-balance implicitly.***

# 5 EXPERIMENTS

## 5.1 EXPERIMENTAL SETUP

### 5.1.1 DATASETS AND LONG-TAILED SETTINGS

We conduct experiments on three benchmark datasets, including CIFAR-10 (Krizhevsky et al., 2009), CIFAR-20 (Krizhevsky et al., 2009) and STL-10 (Coates et al., 2011). Among them, CIFAR-20 is a version of CIFAR-100 that uses 20 super-classes. Since these three datasets are all balanced datasets, we manually generated a total of six long-tailed distributions using two imbalance ratios:

5 and 10. The imbalance ratio, which is the ratio of the maximum and minimum classes, controls the long-tailed degree of data distributions. We generate the sample number for each class following the setting in (Tang et al., 2020; Zhou et al., 2020; Cao et al., 2019), where the number of each class in long-tailed distributions exponentially decreases. We use the train dataset in CIFAR-10 and CIFAR-20, both train and test dataset in STL-10 (due to its small train dataset) for long-tailed deep clustering, following (Tao et al., 2020). Besides, we have also conducted experiments on CIFAR-10 with an imbalance ratio of 100 and on Tiny ImageNet (Le and Yang, 2015) with an imbalance ratio of 10, further demonstrating the effectiveness of ConMix in scenarios with more long-tailed distributions and a greater number of classes. We also report clustering results on ImageNet-LT (Liu et al., 2019) in Appendix F to demonstrate the effectiveness and generalization of ConMix.

### 5.1.2 IMPLEMENTATION DETAILS

We use ResNet-18 (He et al., 2016) as the backbone for all experiments, unless otherwise specified. We train all methods for 1000 epochs and report results of the last epoch for fair comparisons. We use a batch size of 512 for all methods unless otherwise specified. We hope to mixup semantically meaningful representation, thus we train our model only using SDCLR (Jiang et al., 2021) for the first 200 epochs. Then we train our model only with ConMix. We set synthesized representation number $M = 100$ in all experiments, although applying diverse $M$ on different datasets may lead to performance improvements. Following compared methods (Huang et al., 2023; Yu et al., 2023; Tao et al., 2020; Li et al., 2023), we use K-means for final representation clustering. Other specific training settings can be found in the Appendix A.

### 5.2 MAIN RESULTS

In this section, we evaluate the proposed ConMix with both baseline and state-of-the-art deep clustering methods on various benchmarks with different imbalance ratio, including 5 and 10. We compare ConMix with nine methods. SimCLR (Chen et al., 2020), MoCo (He et al., 2020), and BYOL (Grill et al., 2020) are self-supervised learning methods which can learn general representation. SDCLR (Jiang et al., 2021) is a long-tailed self-learning method specially proposed for long-tailed distributions. CC (Li et al., 2021b) is a deep clustering method based on contrastive learning, and it outputs clustering assignment directly. IDFD (Tao et al., 2020), CoNR (Yu et al., 2023), ProPos (Huang et al., 2023) and DMICC (Li et al., 2023) improve better representations for deep clustering.

We evaluate the effectiveness of long-tailed deep clustering using four distinct metrics, including accuracy (ACC), class-averaged accuracy (CAA), normalized mutual information (NMI), adjusted rand index (ARI). Aside from CAA, all are common metrics for evaluating clustering performance. Due to the fact that in long-tailed deep clustering, the same distribution is used for training and testing, accurate head class prediction may lead to undeservedly high ACC. Thus, We introduce CAA (class-averaged accuracy), which is the average accuracy of each class, to compare the performance of different methods fairly. The results of comparison of our method with diverse approaches are shown in Table 1 and 2, and the best result are displayed in bold. In Table 1 and 2, we can observe

Table 1: Clustering results (in percent %) of various methods on three benchmark datasets with imbalance ratio = 5.

| Datasets | CIFAR-10 | | | | CIFAR-20 | | | | STL-10 | | | |
|---|---|---|---|---|---|---|---|---|---|---|---|---|
| Metric | ACC | CAA | NMI | ARI | ACC | CAA | NMI | ARI | ACC | CAA | NMI | ARI |
| SimCLR | 41.4 | 46.1 | 40.5 | 24.6 | 39.7 | 38.7 | 40.4 | 23.8 | 28.6 | 30.3 | 23.9 | 11.8 |
| MoCo | 38.8 | 42.1 | 36.1 | 23.8 | 26.8 | 25.7 | 24.5 | 12.1 | 38.4 | 38.9 | 34.9 | 22.2 |
| BYOL | 51.8 | 52.5 | 52.2 | 34.1 | 35.9 | 34.7 | 36.4 | 21.5 | 39.2 | 40.7 | 33.6 | 22.9 |
| SDCLR | 44.1 | 50.5 | 43.4 | 38.0 | 40.2 | 38.7 | 40.3 | 23.8 | 35.8 | 37.4 | 34.4 | 19.3 |
| CC | 25.2 | 20.9 | 18.3 | 1.47 | 16.0 | 12.0 | 18.0 | 00.0 | 41.5 | 30.0 | 46.2 | 22.4 |
| IDFD | 56.7 | 63.0 | 51.8 | 37.1 | 31.2 | 30.7 | 30.5 | 16.2 | 41.6 | 41.2 | 39.6 | 24.3 |
| CoNR | 41.4 | 43.5 | 34.9 | 23.3 | 23.8 | 23.4 | 21.6 | 12.1 | 32.4 | 31.3 | 29.0 | 18.2 |
| ProPos | 51.4 | 59.2 | 52.2 | 34.1 | 40.7 | 39.3 | 42.6 | 26.4 | 37.2 | 39.3 | 34.5 | 21.8 |
| DMICC | 40.6 | 42.5 | 36.9 | 25.4 | 25.2 | 23.1 | 20.6 | 7.5 | 45.9 | 45.6 | 45.3 | 35.5 |
| ConMix | **61.6** | **65.4** | **59.8** | **45.6** | **42.8** | **41.0** | **43.9** | **27.7** | **48.9** | **49.7** | **48.8** | **35.9** |

that ConMix achieves comprehensively superior performance compared to other comparative methods. General representation learning methods do not necessarily yield inferior results compared to

Table 2: Clustering results (in percent %) of various methods on three benchmark datasets with imbalance ratio = 10.

| Datasets | CIFAR-10 | | | | CIFAR-20 | | | | STL-10 | | | |
|---|---|---|---|---|---|---|---|---|---|---|---|---|
| Metric | ACC | CAA | NMI | ARI | ACC | CAA | NMI | ARI | ACC | CAA | NMI | ARI |
| SimCLR | 39.4 | 42.5 | 38.4 | 23.7 | 34.4 | 33.7 | 36.9 | 19.8 | 27.7 | 28.1 | 22.5 | 11.5 |
| MoCo | 37.0 | 40.8 | 34.7 | 23.0 | 26.7 | 25.0 | 24.0 | 12.1 | 38.1 | 32.8 | 34.8 | 23.7 |
| BYOL | 46.0 | 45.8 | 51.8 | 36.6 | 36.4 | 34.7 | 38.4 | 21.9 | 37.3 | 35.4 | 33.8 | 24.1 |
| SDCLR | 38.9 | 44.3 | 42.5 | 26.5 | 37.8 | 35.9 | 39.6 | 22.9 | 34.6 | 37.9 | 32.2 | 17.3 |
| CC | 40.6 | 27.5 | 43.9 | 18.8 | 19.9 | 14.3 | 21.9 | 1.1 | 43.0 | 35.3 | 44.7 | 25.4 |
| IDFD | 47.5 | 54.9 | 48.4 | 33.1 | 28.7 | 27.2 | 28.6 | 15.1 | 38.6 | 34.7 | 36.8 | 22.8 |
| CoNR | 31.4 | 44.3 | 29.2 | 17.8 | 20.3 | 17.9 | 17.3 | 8.0 | 34.8 | 32.5 | 30.7 | 21.0 |
| ProPos | 46.1 | 49.3 | 52.5 | 34.2 | 36.8 | 33.6 | 40.1 | 22.5 | 35.6 | 38.2 | 37.2 | 23.9 |
| DMICC | 36.6 | 39.5 | 36.8 | 25.9 | 24.7 | 21.9 | 20.7 | 10.1 | 41.3 | 41.9 | 38.7 | 30.3 |
| ConMix | **53.3** | **58.2** | **57.1** | **40.8** | **41.7** | **39.3** | **43.6** | **27.0** | **47.4** | **48.7** | **48.2** | **33.9** |

deep clustering methods, as they merely focus on learning good individual-level representations. Some deep clustering methods may collapse to very poor performance on long-tailed distributions, primarily because the majority of samples are predicted to belong to a few dominant classes (e.g., CoNR and CC on CIFAR-20). Therefore, to apply deep clustering methods effectively to the naturally occurring long-tailed distributions, it is essential to adequately address long-tailed distributions. Moreover, some methods achieve high ACC but not so high ARI (e,g, IDFD on CIFAR-10), which is due to insensitivity to class imbalance, leading to some class being dispersed across multiple clusters or multiple classes being incorrectly grouped into a single cluster.

## 5.3 ABLATION STUDY

We have conducted an extensive series of ablation studies on CIFAR-10 with imbalance ratio of 10 to comprehend the underlying reasons for the experiments' effectiveness, as shown in Table 3. We have conducted an experiment to employ input-level mixup (Zhang et al., 2018) under the SimCLR framework in an unsupervised manner, with all settings consistent with ConMix. We can observe that in the context of long-tailed unsupervised learning, mixup does not enhance the model generalization. In fact, it slightly diminishes the model performance. We think that the model is unable to learn certain original features from the mixed images, which results in a decline in the model representation learning capability. Besides, we conducted unsupervised Manifold Mixup, which interpolates hidden features in the random hidden layer as described in (Verma et al., 2019). We have also conducted pairwise ConMix, where synthesized representations are generated by pairing samples instead of using multi-sample combinations. Note that input-level mixup, unsupervised Manifold Mixup and pairwise ConMix all sample mixing coefficients from beta distributions and utilize a 200-epoch SDCLR warmup. Their experimental results show that both mixup only at the representation level and multi-sampling strategy work effectively. We apply SDCLR to warmup ConMix for 200 epochs to ensure meaningful representations to mixup later. However, the experiments suggest that the effect of this warmup is marginal and does not play a decisive role. It can be inferred that mixup at representation level is more robust compared to input level, because representation-level mixup does not impede the model to learn good representations. Our attempt to substitute SDCLR with SimCLR for warmup also yields desirable results. Note that the method we adopted in experiments is ConMix w/ SDCLR warmup as described in Section 5.1.2.

Table 3: Ablation study (in percent %) of various methods on CIFAR-10 with imbalance ratio = 10.

| Metric | ACC | CAA | NMI | ARI |
|---|---|---|---|---|
| Input-level mixup | 36.8 | 37.8 | 28.3 | 19.5 |
| SimCLR | 39.4 | 42.5 | 38.4 | 23.7 |
| Unsupervised Manifold Mixup | 49.4 | 54.3 | 54.4 | 38.2 |
| Pairwise ConMix | 50.7 | 56.1 | 56.8 | 39.8 |
| ConMix w/o SDCLR warmup | 50.6 | 56.1 | 55.8 | 39.6 |
| ConMix w/ SimCLR warmup | 51.3 | 56.4 | 56.4 | 39.8 |
| ConMix w/ SDCLR warmup | 53.3 | 58.2 | 57.1 | 40.8 |

## 5.4 CLUSTERING IN MORE LONG-TAILED SCENARIOS

We have also conducted experiments under more long-tailed conditions, where ConMix outperforms several approaches as well. We compare ConMix with some representation learning methods and state-of-the-art deep clustering methods when the imbalance ratio reaches 100 on CIFAR-10. However, we have neither explored scenarios with even longer tails nor conducted experiments across multiple datasets at this imbalance ratio, primarily because there remains considerable room for improvement when the imbalance ratio is less severe. In Table 4, the clustering performance on CIFAR-10 with imbalance ratio = 100 is shown. Although the advantages of ConMix over other methods are marginal in terms of ACC and CAA, it still demonstrates significantly improved performance in NMI and ARI. This suggests that the distribution obtained by ConMix is closer to the true underlying distribution.

Table 4: Clustering results (in percent %) of various methods on CIFAR-10 with imbalance ratio = 100.

| Methods | SimCLR | SDCLR | BYOL | CoNR | ProPos | DMICC | ConMix |
|---------|--------|-------|------|------|--------|-------|--------|
| ACC | 32.2 | 32.8 | 33.5 | 28.1 | 38.2 | 31.3 | 40.4 |
| CAA | 31.6 | 32.3 | 35.5 | 20.9 | 38.3 | 26.1 | 38.5 |
| NMI | 36.4 | 39.4 | 43.1 | 25.6 | 42.4 | 41.1 | 53.4 |
| ARI | 20.2 | 22.2 | 25.3 | 14.6 | 24.9 | 22.4 | 33.5 |

## 5.5 CLUSTERING ON THE LARGE-SCALE DATASET

Following (Huang et al., 2023; Yu et al., 2023), we have evaluated on Tiny ImageNet, which contains 200 classes. Since Tiny ImageNet is a balanced dataset, we created a long-tailed version with an imbalance ratio of 10, as described in Section 5.1.1. The results are shown in Table 5. It can be seen that our method still performs better on large-scale long-tailed datasets. Some deep clustering methods perform worse than the baseline methods. We believe this is because these methods assume class balance and perform better under balanced conditions, but do not work as effectively under imbalanced conditions. This also demonstrates the necessity of studying long-tailed deep clustering.

Table 5: Clustering results (in percent %) of various methods on Tiny ImageNet with imbalance ratio = 10.

| Methods | SimCLR | SDCLR | BYOL | CoNR | ProPos | DMICC | ConMix |
|---------|--------|-------|------|------|--------|-------|--------|
| ACC | 16.0 | 14.6 | 16.1 | 8.8 | 15.0 | 9.2 | 16.5 |
| CAA | 14.2 | 12.9 | 14.0 | 8.2 | 13.2 | 8.8 | 14.5 |
| NMI | 33.9 | 35.2 | 34.2 | 28.2 | 34.4 | 28.7 | 34.9 |
| ARI | 7.9 | 8.6 | 8.6 | 3.9 | 7.5 | 3.7 | 8.3 |

## 5.6 CONMIX BENEFITS REPRESENTATION LEARNING

To further investigate the impact of ConMix on representation learning, we test the distributions of representations in each class for CIFAR-10 when the imbalance ratio is 10. We gauge the compactness of class-specific representations by the average similarity within each class, and the separability from other classes by the average dissimilarity between representations within a class and those of other classes. Ideally, the former should be high, indicating a tight grouping of representations within the same class, while the latter should be low, suggesting a clear distinction between representations of different classes. As shown in Figure 1, the proposed ConMix can definitely improve the similarity within the same class and dissimilarity between different classes, jointly indicating that the representations become more discriminative and the biased model has been corrected effectively.

## 5.7 CONMIX WORKS ON UNSEEN BALANCED DATASETS

Experiments above are conducted by training on imbalanced datasets and testing on the same datasets, which is a common practice in deep clustering. Although we employ the CAA to indicate the average clustering performance across different classes, the employed K-means tends to

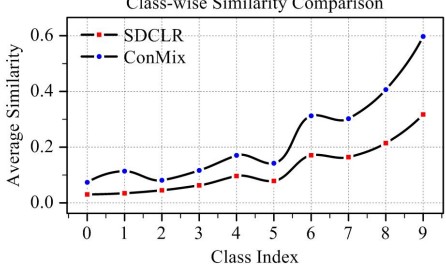 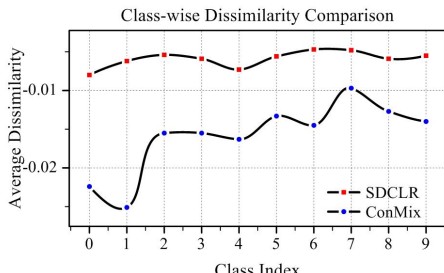

Figure 1: Class-wise similarity and dissimilarity on CIFAR-10 with imbalance ratio = 10. The index of classes ranges from 0 to 9, and the number of samples is decreasing. Left: class-wise similarity indicates the average similarity between all samples within the same class, the higher the better. It can be seen that ConMix can significantly improve this indicator, especially for tail class samples. Thus tail class samples are more compact in high-dimensional space. Right: Class-wise dissimilarity indicates the average similarity between samples within and outside the indexed class, the lower the better. It can be seen that ConMix can significantly lower this indicator, especially for head class samples. It can inferred that head class samples are more discriminative from tail class ones after ConMix.

form clusters based on the overall data distributions. To further examine the practical performance of these biased models trained on long-tailed distributions, we apply models to cluster on balanced datasets. Specifically, we train models on CIFAR-10 with an imbalance ratio of 10 and then evaluate them using the test set of CIFAR-10 as the balanced benchmark. In test set of CIFAR-10, each class has 1,000 samples, which are not present in the train set. Given that the data used for testing is unfamiliar to the models, the clustering results indirectly reflect model generalization, as shown in Table 6. We also train ConMix on a balanced CIFAR-10 dataset, using an equivalent total number of samples as in the long-tailed CIFAR-10 (name as ConMix-B in Table 6). The performance of this balanced model was likewise evaluated on a test set, revealing that the performance drop induced by the long-tailed distribution is approximately 5%, highlighting that ConMix is robust to imbalanced data distributions.

Table 6: Clustering results on balanced test set by models trained on CIFAR-10 with imbalance ratio = 10 or balanced CIFAR-10 with equal number of samples.

| Methods | SimCLR | SDCLR | BYOL | CoNR | ProPos | DMICC | ConMix | ConMix-B |
|---------|--------|-------|------|------|--------|-------|--------|----------|
| ACC | 53.8 | 54.6 | 63.2 | 36.1 | 55.1 | 40.7 | 67.9 | 72.3 |
| NMI | 46.4 | 47.2 | 51.9 | 27.7 | 53.9 | 33.2 | 59.5 | 62.3 |
| ARI | 36.0 | 36.4 | 41.8 | 16.5 | 41.5 | 20.0 | 52.1 | 56.3 |

## 6    CONCLUSION

In this paper, we introduce a novel method for long-tailed deep clustering, aiming at mitigating the bias caused by training models on long-tailed distributions. The proposed method, ConMix, performs multi-sampling linear interpolation at the representation level, effectively extending mixup to deep clustering and contrastive learning. This simple yet effective approach achieves remarkable performance, when compared to state-of-the-art deep clustering methods on several benchmarks datasets. ConMix effectively enhances representation learning, yielding more compact within-class and more separated between-class representations, which are highly conducive to clustering. We also theoretically demonstrate that ConMix implicitly re-balances the loss between head and tail classes in long-tailed learning, thus avoid the common issue in contrastive learning where losses for head classes tend to be larger and those for tail classes tend to be smaller. However, long-tailed deep clustering still lags significantly in performance compared to both long-tailed supervised learning and balanced supervised learning. We posit that there is a need for more research to focus on this largely unexplored realm, in order to advance the development and application of deep clustering.

ACKNOWLEDGMENTS

This work was supported by the National Natural Science Foundation of China under Grant U24A20322, and the Big Data Computing Center of Southeast University.

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

## A    TRAINING SPECIFICS

Here we provide the training specifics that are omitted in the main text due to space limitations. Following (Chen et al., 2020), we set the first convolution layer with kernel size 3×3 and stride 1, and remove the first max-pooling layer for all experiments on CIFAR-10 and CIFAR-20 due to their small image sizes. For ConMix, we adopt the stochastic gradient descent (SGD) optimizer, whose learning rate is 0.5, weight decay is 0.0001 and momentum is 0.9. We adopt the cosine decay learning rate schedule to update the learning rate by step, with 10 epochs for learning rate warmup. The temperature $\tau$ in NT-Xent is 0.2. We adopt the data augmentation methods described in (Chen et al., 2020).

To avoid the uncertainty caused by K-means random initialization, all methods evaluated by K-means will be averaged over 10 trials with different random seeds. Regarding the parameters settings for other comparative methods, we directly adopt the configurations as specified within their respective papers.

Our experiments are based on Pytorch and all models are trained on NVIDIA GeForce RTX 4090 GPUs. It takes approximately five to seven hours on a single GPU to train a model under different long-tailed distributions in the Section 5.2, with the actual time depending on factors such as the size of the dataset.

## B    MULTI-SAMPLE STRATEGY BENEFITS CONMIX

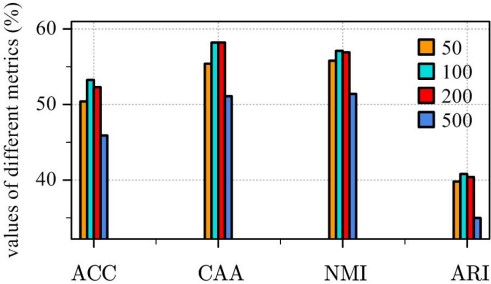

Figure 2: Clustering Results (in percent %) of Different Tag Numbers on CIFAR-10 when Imbalance Ratio = 10.

We demonstrate through ablation studies that the pairwise ConMix remains effective, albeit less so than multi-sample methods. Herein, we further investigate the impact of the tag number $M$ on clustering performance. In our experiments, we employ the long-tailed distribution using CIFAR-10, with an imbalance ratio of 10. We set the tag number $M$ to 50, 100, 200, and 500 respectively. The results indicate that the clustering performance is not significantly influenced by the tag number, unless the multi-sample mixup strategy itself is severely hindered. Due to the batch size of 512, when $M$ is set to 500, the multi-sample approach diminishes considerably, leading to a drop in experimental performance.

## C    DERIVATIONS AND PROOFS OF THE FORMULAS

Due to space limitations, we have omitted the derivation and proof of the formula in the main text. In this section, we will provide detailed derivations and proofs of formulas that appear in the paper.

## C.1 Derivation for $f_{\text{CL}}$

The $\mathcal{L}_{\text{CL}}$ in Eq. (1) can be rewritten into a form of $f_{\text{CL}}$ in Eq. (5):

$$
\begin{aligned}
\mathcal{L}_{\text{CL}} &= \frac{1}{2N} \sum_{i=1}^{2N} -\log \frac{s\left(\boldsymbol{v}_i, \boldsymbol{v}_i^+, \tau\right)}{\sum_{k=1}^{2N} \mathbf{1}_{[k \neq i]} s\left(\boldsymbol{v}_i, \boldsymbol{v}_k, \tau\right)} = \frac{1}{2N} \sum_{i=1}^{2N} \log \frac{\sum_{k=1}^{2N} \mathbf{1}_{[k \neq i]} s\left(\boldsymbol{v}_i, \boldsymbol{v}_k, \tau\right)}{s\left(\boldsymbol{v}_i, \boldsymbol{v}_i^+, \tau\right)} \\
&= \frac{1}{2N} \sum_{i=1}^{2N} \log \sum_{k=1}^{2N} \mathbf{1}_{[k \neq i]} \frac{s\left(\boldsymbol{v}_i, \boldsymbol{v}_k, \tau\right)}{s\left(\boldsymbol{v}_i, \boldsymbol{v}_i^+, \tau\right)} = \frac{1}{2N} \sum_{i=1}^{2N} \log \sum_{k=1}^{2N} \mathbf{1}_{[k \neq i]} \frac{\exp(\boldsymbol{v}_i^\top \boldsymbol{v}_k / \tau)}{\exp(\boldsymbol{v}_i^\top \boldsymbol{v}_i^+ / \tau)} \\
&= \frac{1}{2N} \sum_{i=1}^{2N} \log \sum_{k=1}^{2N} \mathbf{1}_{[k \neq i]} \exp(\boldsymbol{v}_i^\top (\boldsymbol{v}_k - \boldsymbol{v}_i^+) / \tau) \\
&= \frac{1}{2N} \sum_{i=1}^{2N} \log \sum_{k=1}^{2N} \mathbf{1}_{[k \neq i]} \exp(f_{\text{CL}}(\boldsymbol{v}_i, \boldsymbol{v}_k, \boldsymbol{v}_i^+) / \tau), \\
&\text{where } f_{\text{CL}}(\boldsymbol{v}_i, \boldsymbol{v}_k, \boldsymbol{v}_i^+) = \boldsymbol{v}_i^\top (\boldsymbol{v}_k - \boldsymbol{v}_i^+).
\end{aligned}
\tag{9}
$$

## C.2 Derivation for $h_{\text{CM}}$

To measure the contribution of the individual $\boldsymbol{v}_i$ to the total loss $\mathcal{L}_{\text{CM}}$, we propose a new function $h_{\text{CM}}(\boldsymbol{v}_i)$:

$$
\begin{aligned}
h_{\text{CM}}(\boldsymbol{v}_i) &= \sum_{y=1}^{2M} f_{\text{CM}}(\boldsymbol{z}_x, \boldsymbol{z}_y, \boldsymbol{z}_x^+) - \sum_{y=1}^{2M} f_{\text{CM}}(\boldsymbol{z}_x', \boldsymbol{z}_y, \boldsymbol{z}_x'^+) \\
&:= \sum_{y=1}^{2M} \frac{1}{|\mathbb{U}_x|} \sum_{j \in \mathbb{U}_x} \boldsymbol{v}_j^\top \big( \frac{1}{|\mathbb{U}_y|} \sum_{k \in \mathbb{U}_y} \boldsymbol{v}_k - \frac{1}{|\mathbb{U}_x|} \sum_{j \in \mathbb{U}_x^+} \boldsymbol{v}_j^+ \big) - \\
&\qquad \sum_{y=1}^{2M} \frac{1}{|\mathbb{U}_x| - 1} \sum_{j \in \mathbb{U}_x \backslash \{i\}} \boldsymbol{v}_j^\top \big( \frac{1}{|\mathbb{U}_y|} \sum_{k \in \mathbb{U}_y} \boldsymbol{v}_k - \frac{1}{|\mathbb{U}_x| - 1} \sum_{j \in \mathbb{U}_x^+ \backslash \{i^+\}} \boldsymbol{v}_j^+ \big) \\
&:= \sum_{k \notin \mathbb{S}_i \cup \mathbb{S}_i^+} \boldsymbol{v}_i^\top \boldsymbol{v}_k - 2M \sum_{k \in \mathbb{S}_i^+} \boldsymbol{v}_i^\top \boldsymbol{v}_k \\
&= \sum_{k=1}^{2N} \boldsymbol{v}_i^\top \boldsymbol{v}_k - \sum_{k \in \mathbb{S}_i} \boldsymbol{v}_i^\top \boldsymbol{v}_k - \sum_{k \in \mathbb{S}_i^+} \boldsymbol{v}_i^\top \boldsymbol{v}_k - 2M \sum_{k \in \mathbb{S}_i^+} \boldsymbol{v}_i^\top \boldsymbol{v}_k \\
&= \sum_{k=1}^{2N} \boldsymbol{v}_i^\top \boldsymbol{v}_k - \sum_{k \in \mathbb{S}_i} \boldsymbol{v}_i^\top \boldsymbol{v}_k - (2M + 1) \sum_{k \in \mathbb{S}_i^+} \boldsymbol{v}_i^\top \boldsymbol{v}_k,
\end{aligned}
\tag{10}
$$

where $\mathbb{U}_x$, $\mathbb{U}_y$ and $\mathbb{U}_x^+$ denotes the set contains indexes of representations synthesizing $\boldsymbol{z}_x$, $\boldsymbol{z}_y$ and $\boldsymbol{z}_x^+$ respectively. $|\cdot|$ denotes the number of elements in the set. $\boldsymbol{z}_x'$ is synthesized from all representations composed of $\boldsymbol{z}_x$ except $\boldsymbol{v}_i$. $\mathbb{S}_i$ and $\mathbb{S}_i^+$ are the sets contain indexes of original representations from $\boldsymbol{z}_x$ and $\boldsymbol{z}_x^+$ respectively, as $\boldsymbol{v}_i$ synthesizes $\boldsymbol{z}_x$ and $\boldsymbol{v}_i^+$ synthesizes $\boldsymbol{z}_x^+$. := denotes the assignment operator. To simplify the problem, we have excluded the effects of normalization. To align with Eq. (6), we do not take into account the weighted average coefficients needed for each similarity. These changes do not affect the measurement of the impact of individual samples on the overall loss, as they are random factors in the experiments. Instead, these changes make the theory more straightforward to explain.

## C.3 Proof for **Theorem 1**

To prove that $h_{\text{CM}}(\boldsymbol{v}_h) - h_{\text{CM}}(\boldsymbol{v}_t) < h_{\text{CL}}(\boldsymbol{v}_h) - h_{\text{CL}}(\boldsymbol{v}_t)$ for head class representation $\boldsymbol{v}_h$ and tail class representation $\boldsymbol{v}_t$, we first define some constants: $N_{h_{in}}$ is the number of representations

that belongs to the same class as the head class representation $\boldsymbol{v}_h$; $N_{h_{out}}$ is the number of representations that are not of the same class as the head class representation $\boldsymbol{v}_h$; $N_{t_{in}}$ is the number of representations that belongs to the same class as the tail class representation $\boldsymbol{v}_t$; $N_{t_{out}}$ is the number of representations that are not of the same class as the tail class representation $\boldsymbol{v}_t$. Given intra-class similarity $s_w$ and inter-class similarity $s_b$, we can compute the outcome of $h_{\mathrm{CL}}(\boldsymbol{v}_h) - h_{\mathrm{CL}}(\boldsymbol{v}_t)$ as below:

$$
\begin{aligned}
& h_{\mathrm{CL}}(\boldsymbol{v}_h) - h_{\mathrm{CL}}(\boldsymbol{v}_t) \\
=& \sum_{k=1}^{2N} \mathbf{1}_{[k \neq h]}(\boldsymbol{v}_h^\top \boldsymbol{v}_k - \boldsymbol{v}_h^\top \boldsymbol{v}_h^+) - \sum_{k=1}^{2N} \mathbf{1}_{[k \neq t]}(\boldsymbol{v}_t^\top \boldsymbol{v}_k - \boldsymbol{v}_t^\top \boldsymbol{v}_t^+) \\
=& \sum_{k=1}^{2N} (\boldsymbol{v}_h^\top \boldsymbol{v}_k - \boldsymbol{v}_h^\top \boldsymbol{v}_h^+) - \sum_{k=1}^{2N} (\boldsymbol{v}_t^\top \boldsymbol{v}_k - \boldsymbol{v}_t^\top \boldsymbol{v}_t^+) - (\boldsymbol{v}_h^\top \boldsymbol{v}_h - \boldsymbol{v}_h^\top \boldsymbol{v}_h^+) + (\boldsymbol{v}_t^\top \boldsymbol{v}_t - \boldsymbol{v}_t^\top \boldsymbol{v}_t^+) \\
=& \sum_{k=1}^{2N} (\boldsymbol{v}_h^\top \boldsymbol{v}_k - \boldsymbol{v}_h^\top \boldsymbol{v}_h^+) - \sum_{k=1}^{2N} (\boldsymbol{v}_t^\top \boldsymbol{v}_k - \boldsymbol{v}_t^\top \boldsymbol{v}_t^+) - (1 - s_w) + (1 - s_w) \\
=& \sum_{k=1}^{2N} (\boldsymbol{v}_h^\top \boldsymbol{v}_k - \boldsymbol{v}_h^\top \boldsymbol{v}_h^+) - \sum_{k=1}^{2N} (\boldsymbol{v}_t^\top \boldsymbol{v}_k - \boldsymbol{v}_t^\top \boldsymbol{v}_t^+) \\
=& N_{h_{in}} s_w + N_{h_{out}} s_b - 2N - (N_{t_{in}} s_w + N_{t_{out}} s_b - 2N) \\
=& N_{h_{in}} s_w + N_{h_{out}} s_b - N_{t_{in}} s_w - N_{t_{out}} s_b \\
=& 2N s_w - N_{h_{out}} s_w + N_{h_{out}} s_b - 2N s_w + N_{t_{out}} s_w - N_{t_{out}} s_b \\
=& (N_{t_{out}} - N_{h_{out}}) s_w - (N_{t_{out}} - N_{h_{out}}) s_b \\
=& (N_{t_{out}} - N_{h_{out}})(s_w - s_b).
\end{aligned}
\tag{11}
$$

The $h_{\mathrm{CM}}(\boldsymbol{v}_h) - h_{\mathrm{CM}}(\boldsymbol{v}_t)$ can be derived in a similar manner as below:

$$
\begin{aligned}
& h_{\mathrm{CM}}(\boldsymbol{v}_h) - h_{\mathrm{CM}}(\boldsymbol{v}_t) \\
=& \sum_{k=1}^{2N} \boldsymbol{v}_h^\top \boldsymbol{v}_k - \sum_{k \in \mathbb{S}_h} \boldsymbol{v}_h^\top \boldsymbol{v}_k - (2M+1) \sum_{k \in \mathbb{S}_h^+} \boldsymbol{v}_h^\top \boldsymbol{v}_k - (\sum_{k=1}^{2N} \boldsymbol{v}_t^\top \boldsymbol{v}_k - \sum_{k \in \mathbb{S}_t} \boldsymbol{v}_t^\top \boldsymbol{v}_k - (2M+1) \sum_{k \in \mathbb{S}_t^+} \boldsymbol{v}_t^\top \boldsymbol{v}_k) \\
=& \sum_{k=1}^{2N} \boldsymbol{v}_h^\top \boldsymbol{v}_k - \sum_{k \in \mathbb{S}_h^+} \boldsymbol{v}_h^\top \boldsymbol{v}_k + \boldsymbol{v}_h^\top \boldsymbol{v}_h - \boldsymbol{v}_h^\top \boldsymbol{v}_h^+ - (2M+1) \sum_{k \in \mathbb{S}_h^+} \boldsymbol{v}_h^\top \boldsymbol{v}_k - \\
& (\sum_{k=1}^{2N} \boldsymbol{v}_t^\top \boldsymbol{v}_k - \sum_{k \in \mathbb{S}_t^+} \boldsymbol{v}_t^\top \boldsymbol{v}_k + \boldsymbol{v}_t^\top \boldsymbol{v}_t - \boldsymbol{v}_t^\top \boldsymbol{v}_t^+ - (2M+1) \sum_{k \in \mathbb{S}_t^+} \boldsymbol{v}_t^\top \boldsymbol{v}_k) \\
=& \sum_{k=1}^{2N} \boldsymbol{v}_h^\top \boldsymbol{v}_k - \sum_{k \in \mathbb{S}_h^+} \boldsymbol{v}_h^\top \boldsymbol{v}_k - (2M+1) \sum_{k \in \mathbb{S}_h^+} \boldsymbol{v}_h^\top \boldsymbol{v}_k - (\sum_{k=1}^{2N} \boldsymbol{v}_t^\top \boldsymbol{v}_k - \sum_{k \in \mathbb{S}_t^+} \boldsymbol{v}_t^\top \boldsymbol{v}_k - (2M+1) \sum_{k \in \mathbb{S}_t^+} \boldsymbol{v}_t^\top \boldsymbol{v}_k) \\
=& \sum_{k=1}^{2N} \boldsymbol{v}_h^\top \boldsymbol{v}_k - (2M+2) \sum_{k \in \mathbb{S}_h^+} \boldsymbol{v}_h^\top \boldsymbol{v}_k - (\sum_{k=1}^{2N} \boldsymbol{v}_t^\top \boldsymbol{v}_k - (2M+2) \sum_{k \in \mathbb{S}_t^+} \boldsymbol{v}_t^\top \boldsymbol{v}_k) \\
=& \sum_{k=1}^{2N} \boldsymbol{v}_h^\top \boldsymbol{v}_k - \sum_{k=1}^{2N} \boldsymbol{v}_t^\top \boldsymbol{v}_k - (2M+2) \sum_{k \in \mathbb{S}_h^+} \boldsymbol{v}_h^\top \boldsymbol{v}_k + (2M+2) \sum_{k \in \mathbb{S}_t^+} \boldsymbol{v}_t^\top \boldsymbol{v}_k.
\end{aligned}
\tag{12}
$$

Note that in the above Eq. (12), we have made equivalent substitutions: $\sum_{k \in \mathbb{S}_h} \boldsymbol{v}_h^\top \boldsymbol{v}_k$ with $\sum_{k \in \mathbb{S}_h^+} \boldsymbol{v}_h^\top \boldsymbol{v}_k$ and $\sum_{k \in \mathbb{S}_t} \boldsymbol{v}_t^\top \boldsymbol{v}_k$ with $\sum_{k \in \mathbb{S}_t^+} \boldsymbol{v}_t^\top \boldsymbol{v}_k$. And from Eq. (11), we can derive that $\sum_{k=1}^{2N} \boldsymbol{v}_h^\top \boldsymbol{v}_k - \sum_{k=1}^{2N} \boldsymbol{v}_t^\top \boldsymbol{v}_k = (N_{t_{out}} - N_{h_{out}})(s_w - s_b)$. So we can further connect $h_{\mathrm{CM}}(\boldsymbol{v}_h) -$

$h_{\text{CM}}(\boldsymbol{v}_t)$ and $h_{\text{CL}}(\boldsymbol{v}_h) - h_{\text{CL}}(\boldsymbol{v}_t)$ as below:

$$
\begin{aligned}
&h_{\text{CM}}(\boldsymbol{v}_h) - h_{\text{CM}}(\boldsymbol{v}_t) \\
=&(N_{t_{out}} - N_{h_{out}})(s_w - s_b) - (2M+2)\sum_{k\in\mathbb{S}_h^+}\boldsymbol{v}_h^\top\boldsymbol{v}_k + (2M+2)\sum_{k\in\mathbb{S}_t^+}\boldsymbol{v}_t^\top\boldsymbol{v}_k \\
=&h_{\text{CL}}(\boldsymbol{v}_h) - h_{\text{CL}}(\boldsymbol{v}_t) + (2M+2)\sum_{k\in\mathbb{S}_t^+}\boldsymbol{v}_t^\top\boldsymbol{v}_k - (2M+2)\sum_{k\in\mathbb{S}_h^+}\boldsymbol{v}_h^\top\boldsymbol{v}_k.
\end{aligned}
\tag{13}
$$

As the number of original representations composing each synthesized representation is roughly equal, and their class distribution aligns with the overall class distribution in the assumption, $\mathbb{S}_h$ usually has more representations within the same class as $\boldsymbol{v}_h$ than $\mathbb{S}_t$ has within the same class as $\boldsymbol{v}_t$. Thus, $(2M+2)\sum_{k\in\mathbb{S}_t}\boldsymbol{v}_t^\top\boldsymbol{v}_k - (2M+2)\sum_{k\in\mathbb{S}_h}\boldsymbol{v}_h^\top\boldsymbol{v}_k < 0$. Along with Eq. (13), we can conclude that

$$
h_{\text{CM}}(\boldsymbol{v}_h) - h_{\text{CM}}(\boldsymbol{v}_t) < h_{\text{CL}}(\boldsymbol{v}_h) - h_{\text{CL}}(\boldsymbol{v}_t).
\tag{14}
$$

Then the **Theorem 1** is proven and ConMix is proven theoretically to relatively reduce the loss of head class samples and increase the loss of tail class samples, thus achieving loss re-balance.

## D    CLUSTERING RESULTS ON BALANCED DATASETS

We further trained and tested the performance of different methods on balanced datasets. The dataset configurations and experimental setups refer to (Huang et al., 2023). Specifically, we trained different methods using ResNet-18 for 1000 epochs on CIFAR-10 and CIFAR-20. Since ConMix does not leverage some advanced techniques suitable for balanced clustering, we propose an updated version of ConMix called "ConMix+Propos" that embeds ConMix into Propos (Huang et al., 2023). We first train the model with the loss of ConMix for 500 epochs, then train it with Propos for another 500 epochs. The total number of training epochs for this updated version is the same as other methods.

We have reported the results of the balanced dataset in Table 7. We have also provided the clustering performance on the datasets with an imbalance ratio of 10 for better comparisons.

Table 7: Clustering results (in percent %) of various methods on balanced datasets and imbalance datasets with an imbalance ratio = 10.

| Datasets | CIFAR-10 | | | | | | CIFAR-20 | | | | | |
|---|---|---|---|---|---|---|---|---|---|---|---|---|
| Data Type | Balanced | | | Imbalanced | | | Balanced | | | Imbalanced | | |
| Metric | ACC | NMI | ARI | ACC | NMI | ARI | ACC | NMI | ARI | ACC | NMI | ARI |
| SimCLR | 72.8 | 63.9 | 56.7 | 39.4 | 38.4 | 23.7 | 45.4 | 43.8 | 28.8 | 34.4 | 36.9 | 19.8 |
| SDCLR | 71.4 | 62.4 | 54.8 | 38.9 | 42.5 | 26.5 | 44.6 | 43.3 | 27.6 | 37.8 | 39.6 | 22.9 |
| CC | 79.0 | 70.5 | 63.7 | 40.6 | 43.9 | 18.8 | 42.9 | 43.1 | 26.6 | 19.9 | 21.9 | 1.1 |
| IDFD | 81.5 | 71.1 | 66.3 | 47.5 | 48.4 | 33.1 | 42.5 | 42.6 | 26.4 | 28.7 | 28.6 | 15.1 |
| Propos | 91.6 | 85.1 | **83.5** | 46.1 | 52.5 | 34.2 | 57.8 | 58.2 | 42.3 | 36.8 | 40.1 | 22.5 |
| ConMix | 80.9 | 70.7 | 65.6 | 53.3 | 57.1 | 40.8 | 46.0 | 45.5 | 29.8 | 41.7 | 43.6 | 27.0 |
| ConMix+Propos | **92.0** | **85.3** | 83.3 | **53.8** | **58.8** | **42.8** | **59.2** | **58.4** | **42.5** | **43.8** | **47.4** | **30.3** |

Compared to the baseline methods (SimCLR, SDCLR), ConMix demonstrates significant performance improvements on both the balanced datasets and the long-tailed datasets.

Compared to recent deep clustering methods (CC, IDFD, Propos), ConMix shows improvements on long-tailed datasets but may not perform as effective as Propos on balanced datasets. Note that, ConMix still outperforms CC and IDFD on most cases even on the balanced datasets.

However, the updated version "ConMix+Propos" performs the best on both balanced datasets and long-tailed datasets, showing that adding some recent deep clustering techniques on "ConMix" will further improve its performance and robustness.

Moreover, we also notice that existing deep clustering algorithms perform well on balanced datasets but suffer severe performance degradation on long-tailed datasets. We believe this is due to these methods making assumptions that are aligned with balanced datasets. While they can achieve good laboratory performance, they are less suitable for realistic long-tailed distributions. This further underscores the importance of research on long-tailed deep clustering.

## E   Clustering Results of Different Clustering Techniques

Our primary consideration for using K-means as the clustering method is that most of our compared methods utilize K-means (except for Contrastive Clustering which directly outputs the clustering assignments). So the use of K-means allows us to achieve a fair comparison.

However, we have investigated the impact of different clustering methods on the performance of ConMix. We tried the Gaussian Mixture Model (GMM) (Reynolds et al., 2009) on the learned embedding of ConMix to obtain cluster assignments. We use two methods to initialize GMM: one is to initialize using K-means, and the other is random initialization. We denote these two cases as GMM-k and GMM-r. Besides, we tested the performance of agglomerative clustering (Lukasová, 1979) on ConMix (we refer to the method briefly as AC). The detailed results are in Table 8.

Table 8: Clustering results (in percent %) of different clustering techniques on imbalance datasets with an imbalance ratio = 10.

| Datasets | CIFAR-10 | | | | CIFAR-20 | | | | STL-10 | | | |
|---|---|---|---|---|---|---|---|---|---|---|---|---|
| Metric | ACC | CAA | NMI | ARI | ACC | CAA | NMI | ARI | ACC | CAA | NMI | ARI |
| K-means | 53.3 | 58.2 | 57.1 | 40.8 | **41.7** | **39.3** | **43.6** | **27.0** | 47.4 | 48.4 | 48.2 | 33.9 |
| GMM-k | **63.6** | 48.1 | **59.1** | **53.1** | 37.2 | 35.4 | 42.7 | 23.7 | **50.6** | 40.7 | 47.3 | **34.1** |
| GMM-r | 50.3 | 49.0 | 58.7 | 44.7 | 31.3 | 22.2 | 34.9 | 18.1 | 45.9 | 46.0 | 46.2 | 31.5 |
| AC | 59.8 | **62.2** | 58.8 | 46.5 | 39.0 | 37.0 | 42.0 | 20.0 | 47.4 | **51.4** | **49.7** | 33.5 |

Compared with these methods, we can find that different clustering methods can all lead to good performance, validating the effectiveness of our method ConMix. However, for different datasets, the methods achieving the best performance may vary. For example, on CIFAR-10, GMM-k achieved the highest ACC, NMI and ARI, while AC achieved the highest CAA. On CIFAR-20, K-means performed the best. On STL-10, GMM-k obtained the best results in terms of ACC and ARI, while AC achieved the highest CAA and NMI.

## F   Clustering results on ImageNet-LT

To further validate the effectiveness and generalization capability of ConMix, we conducted experiments on the long-tailed dataset ImageNet-LT (Liu et al., 2019). It is the long-tailed subset of ImageNet-1K (Deng et al., 2009), consisting of 115.8K images spanning 1,000 classes, with sample number ranging from 1280 to 5. Following (Li et al., 2021a), we trained a ResNet-50 for 200 epochs and reported the results of the last epoch. We compare ConMix with three baseline methods (Sim-CLR, SDCLR, BYOL) and three recent superior deep clustering methods (IDFD, CoNR, DMICC). The results are in the Table 9. The experimental results demonstrate that our method still performs

Table 9: Clustering results (in percent %) of various methods on ImageNet-LT.

| | SimCLR | SDCLR | BYOL | IDFD | CoNR | DMICC | ConMix |
|---|---|---|---|---|---|---|---|
| ACC | 14.7 | 13.7 | 14.8 | 4.42 | 6.38 | 5.24 | **15.4** |
| CAA | 11.3 | 10.5 | 10.9 | 5.58 | 6.85 | 5.94 | **12.2** |
| NMI | 51.4 | 50.3 | 50.6 | 35.6 | 38.9 | 37.9 | **51.6** |
| ARI | 9.14 | 9.30 | 10.2 | 1.17 | 2.19 | 1.83 | **11.4** |

well on ImageNet-LT, proving its effectiveness and generalization capability. At the same time, It may be surprising that recent state-of-the-art methods perform worse than the baseline methods. The reason is that they make assumption that data are balanced distributed, which conflicts with the real distribution of the data. This phenomenon indicates the limitations of current deep clustering methods in handling long-tailed data and highlights the urgent need for research into long-tailed deep clustering.

## G   Empirical Analysis on the Impact of ConMix

We conducted experiments on CIFAR-10 with imbalance ratios (IR) of 1, 2, 5, 10, 20, 50, and 100. And we provide the empirical analysis from the perspective of the compactness of the learned

representations for different classes. Specifically, we train the baseline SimCLR and ConMix on CIFAR-10 under different imbalance ratios and calculate the class-wise similarity for each class. Class-wise similarity denotes the average similarity of samples within the same class and can indicate the compactness of representations of different classes in the feature space. Thus, to facilitate the description, we will refer to this metric as "compactness" below. We categorize the 10 classes of CIFAR-10 into Many, Medium, and Few categories based on the number of samples, following a 3:4:3 ratio. We first calculate the compactness for each class, then compute the average compactness for the three categories.

Moreover, we propose a new metric, denoted as F/My, which represents the ratio of the Few category compactness to the Many's, to measure the balance between head classes (Many) and tail classes (Few). The larger F/My is, the greater the discrepancy in compactness between head classes and tail classes is, indicating a more severe impact of the long-tailed distribution. The results are shown in the Table 10.

Table 10: The comparison of compactness and F/My on CIFAR-10 with different imbalance ratios.

|  | Method | Many | Medium | Few | F/My |
|---|---|---|---|---|---|
| IR=1 | SimCLR | 0.08 | 0.08 | 0.11 | 1.39 |
| | ConMix | 0.16 | 0.16 | 0.20 | 1.25 |
| IR=2 | SimCLR | 0.05 | 0.08 | 0.11 | 2.25 |
| | ConMix | 0.13 | 0.16 | 0.26 | 1.98 |
| IR=5 | SimCLR | 0.03 | 0.08 | 0.16 | 4.74 |
| | ConMix | 0.10 | 0.17 | 0.35 | 3.36 |
| IR=10 | SimCLR | 0.03 | 0.08 | 0.22 | 7.74 |
| | ConMix | 0.08 | 0.18 | 0.43 | 4.85 |
| IR=20 | SimCLR | 0.02 | 0.09 | 0.26 | 10.36 |
| | ConMix | 0.08 | 0.20 | 0.46 | 5.69 |
| IR=50 | SimCLR | 0.02 | 0.12 | 0.27 | 13.11 |
| | ConMix | 0.12 | 0.27 | 0.36 | 5.08 |
| IR=100 | SimCLR | 0.02 | 0.13 | 0.25 | 12.55 |
| | ConMix | 0.07 | 0.26 | 0.30 | 4.22 |

From the table, we can derive the following empirical analysis:

(1) The head classes (Many) typically have smaller compactness values, while tail classes (Few) have larger compactness values. This suggests that due to long-tailed effect, head classes occupy more of the feature space than tail classes.

(2) When the distribution is long-tailed, the compactness for Few category is relatively higher, while the compactness for Many category is relatively lower, leading to a larger F/My ratio. This is a negative effect of the long-tailed distribution. However, ConMix can reduce the F/My ratio compared to SimCLR under different imbalance ratios, indicating its ability to mitigate the impact of the long-tailed distributions.

(3) Regardless of whether the classes are Few, Medium, or Many, ConMix improves compactness across different imbalance ratios. This indicates that samples within the same class become more compact, which is beneficial for clustering.

(4) When the imbalance ratio is 1, the dataset is balanced and differences in compactness across different classes are due to the varying difficulty of learning each class.

The above analysis empirically demonstrates that ConMix benefits long-tailed clustering.

