# OpenReview forum: "ConMix: Contrastive Mixup at Representation Level for Long-tailed Deep Clustering"
_ICLR.cc/2025/Conference — ICLR 2025 Poster_

### Official Review · Reviewer_RUUY · 2024-11-02

**Soundness:** 2
**Presentation:** 2
**Contribution:** 2
**Rating:** 6
**Confidence:** 3

**Summary:**

The paper proposes to leverage mixup to improve deep clustering approaches for imbalanced datasets. In particular, a multi-sample mixup is incorporated into the SimCLR loss. Instead of just contrasting two augmentations of the same sample, a random subset of samples are selected and the mean representations of their two augmentations are contrasted. A theoretical analysis is performed that shows, under a certain set of simplifications, that this procedure increases the loss of the underrepresented classes. Further, empirical evaluation demonstrate that the scheme can outperform alternative approaches in the imbalanced setting.

**Strengths:**

Limited work has been done on considering imbalance in deep clustering, with most approaches adopting a balanced assumption, and approaches addressing this shortcomings are thus of significance to the community.

While mixup on a representations has previously been integrated into contrastive learning (also mixing multiple samples), there is a certain novelty of leveraging this in the clustering setting to address class imbalance, which is further supported by the theoretical analysis.

The proposed approach is simple and appears to be effective in the settings considered in this work.

**Weaknesses:**

While the author’s main focus is on the imbalanced setting, it would be beneficial to also include comparisons in a balanced setting to be able to judge the overall ability of the method.

The overall clarity of Section 3.3. can be improved. How are the “stochastically assigned tags” selected? Does each sample have a certain probability of being included (independent of each other)? If that is the case, what is the probability set to? Also, in line 215, the notation of the cardinality of the set is not aligned with Eq. 3.

The comparison to pairwise ConMix (standard manifold mixup) in Table 3 is not clear. It appears that the pairwise mixup obtains equivalent results to ConMix w/o SDCLR warmup and it is unclear if pairwise ConMix leverages SDCLR warmup here. Also, is this pairwise ConMix directly leveraging the mean representation of the pair or do the authors create a convex combination with weights sampled from a beta distribution?

As deep clustering methods tend to be a bit less stable than supervised models, some measures of variability/statistical significance would be beneficial in Table 3.

**Questions:**

Could the authors elaborate on the statement in Line 220: “…. Equivalent to implicitly sampling different weights from the beta distribution”. Do the authors refer to the mixing coefficient in the original mixup formulation? While the reviewer understands that the cardinality of the set U_m follows a beta distribution, the final mixup representation will be the mean representations of these samples, which is different from the mixing coefficients.

Further, could the authors comment on the performance of the proposed approach in the balanced setting and on the results in Table 3 with regards to the pairwise mixup and the variability of the results?

---

> ### Author Response · Authors · 2024-11-21
> **Response to Reviewer RUUY (1/2)**
>
> ## Response to Reviewer RUUY (1/2)
> Thank you for reading our paper carefully. We have carefully prepared our response as follows.
>
> > ### Weakness 1 and Question 2: Could the authors comment on the performance of the proposed approach in the balanced setting.
>
> **A**: Thank you for this valuable suggestion. Following your suggestion, we further trained and tested the performance of different methods on balanced datasets. The dataset configurations and experimental setups refer to [1]. Specifically, we trained different methods using ResNet-18 for 1000 epochs on CIFAR-10 and CIFAR-20. Since ConMix does not leverage some advanced techniques suitable for balanced clustering, we propose an updated version of ConMix called "ConMix+Propos" that embeds ConMix into Propos [1]. We first train the model with the loss of ConMix for 500 epochs, then train it with Propos for another 500 epochs. The total number of training epochs for this updated version is the same as other methods.
>
> We have reported the results of the balanced dataset in the following tables, and also provided the clustering performance on the datasets with an imbalance ratio of 10 in the **parentheses** ( $\cdot$ ), for better comparisons.
>
> The experimental results are as follows.
>
> CIFAR-10:
>
> |         | ACC         | NMI         | ARI         |
> |---------|-------------|-------------|-------------|
> | SimCLR  | 72.8 (39.4) | 63.9 (38.4) | 56.7 (23.7) |
> | SDCLR   | 71.4 (38.9) | 62.4 (42.5) | 54.8 (26.5) |
> | CC      | 79.0 (40.6) | 70.5 (43.9) | 63.7 (18.8) |
> | IDFD    | 81.5 (47.5) | 71.1 (48.4) | 66.3 (33.1) |
> | Propos  | 91.6 (46.1) | 85.1 (52.5) | **83.5** (34.2) |
> | ConMix  | 80.9 (53.3) | 70.7 (57.1) | 65.6 (40.8) |
> | ConMix+Propos | **92.0** (**53.8**) | **85.3** (**58.8**) | 83.3 (**42.8**) |
>
>
> CIFAR-20:
>
> |         | ACC         | NMI         | ARI         |
> |---------|-------------|-------------|-------------|
> | SimCLR  | 45.4 (34.4) | 43.8 (36.9) | 28.8 (19.8) |
> | SDCLR   | 44.6 (37.8) | 43.3 (39.6) | 27.6 (22.9) |
> | CC      | 42.9 (19.9) | 43.1 (21.9) | 26.6 ( 1.1) |
> | IDFD    | 42.5 (28.7) | 42.6 (28.6) | 26.4 (15.1) |
> | Propos  | 57.8 (36.8) | 58.2 (40.1) | 42.3 (22.5) |
> | ConMix  | 46.0 (41.7) | 45.5 (43.6) | 29.8 (27.0) |
> | ConMix+Propos | **59.2** (**43.8**) | **58.4** (**47.4**) | **42.5** (**30.3**) |
>
> Compared to the baseline methods (SimCLR, SDCLR), ConMix **demonstrates significant performance improvements on both the balanced datasets and the long-tailed datasets.** Compared to recent deep clustering methods (CC, IDFD, Propos), ConMix shows improvements on long-tailed datasets but may not perform as effective as Propos on balanced datasets. Note that, ConMix still outperforms CC and IDFD on most cases even on the balanced datasets.
>
> However, the updated version "ConMix+Propos" performs the best on both balanced datasets and long-tailed datasets, showing that adding some recent deep clustering techniques on "ConMix" will further improve its performance and robustness.
>
> Moreover, we also notice that existing deep clustering algorithms perform well on balanced datasets **but suffer severe performance degradation on long-tailed datasets.** We believe this is due to these methods making assumptions that are aligned with balanced datasets. While they can achieve good laboratory performance, they are less suitable for realistic long-tailed distributions. This further underscores the importance of research on long-tailed deep clustering.
>
> The above results and analyses have been added in Appendix C of the updated paper to demonstrate robustness of our work. Thank you.
>
> [1] Learning Representation for Clustering Via Prototype Scattering and Positive Sampling, TPAMI, 2023.
>
> >### Weakness 2: How are the "stochastically assigned tags" selected?
>
> **A**: In each batch, we assign a random tag to each input for synthesized representations generation. For example, when the batch size is 512 and the number of synthesized representations $M$ is 100, we generate a sequence of 512 random numbers, with values ranging from 1 to 100, where the generation of these random tags follows a uniform distribution, i.e., for a single input, the probability of being assigned to any specific tag is $\frac{1}{M}$.
>
> In the revised paper, we have  modified Line 216 from "We randomly select...with equal contributions" to:
>
> "In each batch, we randomly assign tags within $[1, M]$ to original representations from the same network branch {$v_1, v_2, ...v_N$}. The generation of tags follows a uniform distribution with equal probabilities $\frac{1}{M}$ and original representations with the same tag are used to synthesize one particular representation in the manner of Eq. (3)."
>
> Also, we apologize for the typo in Line 215. $\vert \cdot\vert$ denotes the number of elements in the set. We have fixed it in the revised version. Thank you very much for your kind reminder.

---

> ### Author Response · Authors · 2024-11-21
> **Response to Reviewer RUUY (2/2)**
>
> ## Response to Reviewer RUUY (2/2)
> >### Weakness 3: Confusion about the settings on pairwise ConMix.
>
> **A**: Sorry for the confusion caused. To control for a single variable in order to verify the improvement of multi-sampling methods over the pairwise method, pairwise ConMix also utilizes a 200-epoch SDCLR warmup. Additionally, pairwise ConMix samples weights from a beta distribution rather than using mean representations. We have clarified these points in updated paper and modified Line 408-409 from "We have also...multi-sample combinations" to：
>
> "We have also conducted pairwise ConMix, where synthesized representations are generated by pairing samples instead of using multi-sample combinations. It samples mixing coefficient from a beta distribution and also utilizes a 200-epoch SDCLR warmup."
>
> We hope this revision can improve readability. Thank you!
>
> >### Weakness 4: Some measures of variability/statistical significance for Table 3.
>
> **A**:  Thank you for your valuable suggestion. Following your advice, we provide the standard deviation and significance test results. Specifically, as the ablation study in the paper, we conduct experiments on CIFAR-10 with an imbalance ratio of 10.  We performed the method 10 times and perform a t-test for significance at the 5% level to compare the results of the proposed method with those of ablation study. ✝ denotes rejection of the original hypothesis and the two results are significantly different. The results in percentage are shown below.
>
> | Metric                  | ACC           | CAA           | NMI           | ARI           |
> |-------------------------|---------------|---------------|---------------|---------------|
> | Input-level mixup       | 36.8±0.54✝    | 37.8±0.19✝    | 28.3±0.43✝    | 19.5±0.35✝    |
> | SimCLR                  | 39.4±2.31✝    | 42.5±4.63✝    | 38.4±1.22✝    | 23.7±1.36✝    |
> | Pairwise ConMix         | 50.7±1.47✝    | 56.1±3.00✝    | 56.8±0.73     | 39.8±0.82✝    |
> | ConMix w/o warmup       | 50.6±2.88✝    | 56.1±4.27     | 55.8±1.28✝    | 39.6±1.88     |
> | ConMix w/ SimCLR warmup | 51.3±1.48✝    | 56.4±2.57✝    | 56.4±0.66✝    | 39.8±0.64✝    |
> | ConMix w/ SDCLR warmup  | **53.3±1.29** | **58.2±0.65** | **57.1±0.78** | **40.8±1.03** |
>
> It can be seen that the standard deviations of the ConMix methods are all within 3%, except for the CAA of ConMix w/o warmup, suggesting the robustness of our method. And the adopted ConMix w/ SDCLR warmup  significantly outperforms other variants on most metrics.
>
> >### Question 1: Confusion about Line 220 "...equivalent to implicitly sampling different weights from the beta distribution".
>
> **A**: We apologize that our explanation may have caused you misunderstanding. In mixup, different weights are sampled from a beta distribution. The multi-sampling strategy of ConMix can also achieve varying weights due to the different numbers of samples with specific tags. For instance, let us assume that there are 3 samples associated with tag 1, and 5 samples associated with tag 2. When obtaining the synthesized representation through averaging, the sets corresponding to these two tags have different cardinalities, leading to different weights for the certain samples involved in the synthesis. Our intention is not to claim that the multi-sampling strategy of ConMix and the weight sampling from a beta distribution in mixup are mathematically identical. We have revised the original sentence to:
>
> "but it also assigns different weights to different samples, similar to how mixup does."
>
> Above is our response. We are grateful for your careful review. We believe that your advice improves our work. We hope our response can be satisfactory. Thank you very much!

---

> ### Author Response · Authors · 2024-11-25
>
> Dear reviewer RUUY,
>
> Thanks again for your time and efforts in reviewing this paper and the valuable comments on improving its quality. As the reviewer-author discussion deadline approaches, we hope to hear your feedback about our response. If you have further concerns, we are happy to provide more explanations. Thank you very much!
>
> Regards from the authors.

---

> > ### Comment · Reviewer_RUUY · 2024-11-25
> >
> > Thank you for the detailed response to my questions and additional clarifications.
> >
> > I have a follow-up question for weakness 3 (the pairwise ConMix). By computing the mean of multiple (randomly selected) samples, the resulting z embeddings will cover a much broader area of the representation space than the mean of pairs of samples. In Manifold Mixup, this is addressed by instead considering the convex combination of pairs with weights sampled from a beta distribution. Could the authors provide a direct comparison in Table 3 to show that the same performance cannot be reached by employing directly manifold mixup instead?

---

> > > ### Author Response · Authors · 2024-11-25
> > > **Response to the follow-up question**
> > >
> > > Thanks for your valuable comments. We will first illustrate the differences between ConMix and pairwise ConMix, then discuss the relationship between pairwise ConMix and Manifold Mixup [1], and finally show the experimental comparisons.
> > >
> > > First, ConMix employs a multi-sampling strategy, while pairwise ConMix is a degraded version of ConMix that only uses a pairwise-sampling strategy. But, we also need to point out that the mixing coefficients of pairwise ConMix follow a beta distribution just like Manifold Mixup [1]. So it is not simply the mean representation of paired samples.
> > >
> > > Second, pairwise ConMix is unsupervised and only performs mixup at the representation level. However, Manifold Mixup is supervised and interpolates hidden features (including representation level features) and labels in a pairwise manner. If we extend the Manifold Mixup to the unsupervised manner and only consider the representation level mixup, the Manfiold mixup will be identical to the pairwise ConMix, and the results in Table 3 show that the proposed ConMix will outperform the Manifold Mixup.
> > >
> > > Finally, the Manifold Mixup can also take the hidden features as input. So, to comprehensively compare the Manifold Mixup with the proposed ConMix, we mixed hidden features in the random hidden layer as described in [1] and continued with forward propagation. Then, we used the NT-Xent loss from SimCLR [2] on the representations output by the network. The experimental results are shown in the table below.
> > >
> > > | Metric                      | ACC        | CAA        | NMI        | ARI        |
> > > |-----------------------------|------------|------------|------------|------------|
> > > | Input-level mixup           | 36.8±0.54✝ | 37.8±0.19✝ | 28.3±0.43✝ | 19.5±0.35✝ |
> > > | SimCLR                      | 39.4±2.31✝ | 42.5±4.63✝ | 38.4±1.22✝ | 23.7±1.36✝ |
> > > | Unsupervised Manifold Mixup | 49.4±1.97✝ | 54.3±3.81✝ | 54.4±0.82✝ | 38.2±0.64✝ |
> > > | Pairwise ConMix             | 50.7±1.47✝ | 56.1±3.00✝ | 56.8±0.73  | 39.8±0.82✝ |
> > > | ConMix w/o warmup           | 50.6±2.88✝ | 56.1±4.27  | 55.8±1.28✝ | 39.6±1.88  |
> > > | ConMix w/ SimCLR warmup     | 51.3±1.48✝ | 56.4±2.57✝ | 56.4±0.66✝ | 39.8±0.64✝ |
> > > | ConMix w/ SDCLR warmup      | 53.3±1.29  | 58.2±0.65  | 57.1±0.78  | 40.8±1.03  |
> > >
> > > The results demonstrate that the performance of our method is clearly better than that of unsupervised Manifold Mixup. Moreover, due to that hidden features often only capture low-level information, interpolating the final output representations of the network yields better results than interpolating hidden features.
> > >
> > > The experimental results have been added to Section 5.3 of the revised paper.
> > >
> > > We hope that the above response can satisfy you. If so, we would appreciate it if you could improve our rating, which is very important to us. If not, please feel free to ask further questions, and we will do our best to meet your needs. Thank you!
> > >
> > > [1] Manifold mixup: Better representations by interpolating hidden states, ICML, 2019.
> > >
> > > [2] A Simple Framework for Contrastive Learning of Visual Representation, ICML, 2020.

---

> ### Author Response · Authors · 2024-11-26
> **Looking forward to your valuable reply**
>
> Dear Reviewer RUUY,
>
> We are very grateful for your willingness to participate in the discussion. Would you please let us know if our latest response has met your satisfaction? Given that the deadline for revising the paper is approaching, with less than two days remaining, we hope to learn whether you have any further concerns. If you have any concerns, we will certainly provide a prompt response and make the corresponding revisions to the paper to improve it. We look forward to your reply with great anticipation. We would be very grateful for your reply amidst your busy schedule. Thank you for the time and effort you have invested in our work.
>
> Regards from the authors.

---

> > ### Comment · Reviewer_RUUY · 2024-11-27
> >
> > Thank you for your additional clarifications, which have addressed most of my concerns. I have raised my score.

---

> > > ### Author Response · Authors · 2024-11-27
> > >
> > > Dear Reviewer RUUY,
> > >
> > > We are grateful for your raising score.
> > > Thank you very much for your careful review and constructive comments.
> > > It is a great honor that you recognize our work.
> > > Wish you everything goes well.
> > > Thank you very much!
> > >
> > > Regards from the authors.

---

### Official Review · Reviewer_EuZq · 2024-11-04

**Soundness:** 3
**Presentation:** 2
**Contribution:** 2
**Rating:** 6
**Confidence:** 3

**Summary:**

The paper presents a novel method, ConMix, aimed at addressing the challenges of long-tailed distributions in deep clustering. The authors argue that existing deep clustering approaches typically assume balanced class distributions, which is not the case in many real-world datasets. ConMix leverages a contrastive mixup strategy to enhance representation learning, theoretically proving its effectiveness in rebalancing class losses without the need for label information. The method is evaluated on benchmark datasets, demonstrating superior performance over existing state-of-the-art approaches.

**Strengths:**

The introduction of ConMix as a contrastive mixup method specifically designed for long-tailed deep clustering is a notable contribution to the field. The approach is innovative, extending mixup techniques into the realm of unsupervised learning.

The authors provide a theoretical foundation for their method, demonstrating how it can implicitly balance losses across head and tail classes. This theoretical insight is valuable and adds depth to the paper.

The evaluations on various benchmark datasets and the assertion of outperforming existing methods lend credibility to the proposed approach. The performance metrics presented seem robust.

**Weaknesses:**

Diversity of Datasets: The experiments are limited to a few benchmark datasets, lacking validation of the method’s effectiveness on more complex and diverse datasets. It is recommended to conduct experiments on larger image classification datasets such as ImageNet to thoroughly evaluate the model’s generalization ability and practicality.

Interpretability of the Method: Although theoretical proofs are provided, the interpretability of how ConMix specifically affects the model learning process remains insufficient. Consider adding comparative experiments to illustrate the specific impacts of ConMix under varying conditions (e.g., different long-tail ratios) to enhance the depth of the paper.

Details of Experimental Setup: The experimental section lacks detailed descriptions of hyperparameter choices and training specifics, which could affect the reproducibility of results. It is suggested to include these details in the methodology section to assist other researchers in understanding and replicating the experiments.

**Questions:**

Conduct additional experiments on large and complex datasets like ImageNet to validate the effectiveness and generalization capability of ConMix.

Enhance the discussion on method interpretability by providing more empirical analysis regarding its impacts.

Provide detailed descriptions of the experimental setup and hyperparameter selections to improve transparency and reproducibility of the research.

---

> ### Author Response · Authors · 2024-11-21
> **Response to Reviewer EuZq (1/3)**
>
> ## Response to Reviewer EuZq (1/3)
> Thank you for your valuable feedback. We have carefully considered your comments and provided our answers below.
>
> > ### Weakness 1 and Question 1: Conduct additional experiments on large and complex datasets like ImageNet to validate the effectiveness and generalization capability of ConMix.
>
> **A**: The adopted datasets are commonly used in the recent deep clustering papers [1-3]. As those recent deep clustering method did not perform experiments on ImageNet-1K, we just followed their settings. We have performed the experiments on Tiny ImageNet in Table 5 of the paper.
>
> However, we agree with your opinion that conducting experiments on a larger dataset to validate the effectiveness and generalization capability of ConMix is important. Therefore, we conducted experiments on the long-tailed dataset ImageNet-LT. It is the long-tailed subset of ImageNet-1K, which consists of 115.8K images spanning 1,000 classes, with sample number ranging from 1280 to 5. Following [4],  we trained a ResNet-50 for 200 epochs and reported the results of the last epoch. We compare ConMix with three baseline methods (SimCLR, SDCLR, BYOL) and three recent superior deep clustering methods (IDFD, CoNR, DMICC).
>
> The results are in the table below.
>
> |        | ACC  | CAA  | NMI  | ARI  |
> |--------|------|------|------|------|
> | SimCLR | 14.7 | 11.3 | 51.4 | 9.14 |
> | SDCLR  | 13.7 | 10.5 | 50.3 | 9.30 |
> | BYOL   | 14.8 | 10.9 | 50.6 | 10.2 |
> | IDFD   | 4.42 | 5.58 | 35.6 | 1.17 |
> | CoNR   | 6.38 | 6.85 | 38.9 | 2.19 |
> | DMICC  | 5.24 | 5.94 | 37.9 | 1.83 |
> | ConMix | **15.4** | **12.2** | **51.6** | **11.4** |
>
> The experimental results demonstrate that our method still performs well on ImageNet-LT, proving its effectiveness and generalization capability. At the same time, It may be surprising that recent state-of-the-art methods perform worse than the baseline methods. The reason is that they make assumption that data are balanced distributed, which conflicts with the real distribution of the data. This phenomenon indicates the limitations of current deep clustering methods in handling long-tailed data and highlights the urgent need for research into long-tailed deep clustering.
>
> The experiments on ImageNet-LT have been added in Appendix E.
>
> [1] Contextually Affinitive Neighborhood Refinery for Deep Clustering, NeurIPS, 2023.
>
> [2] Clustering-Friendly Representation Learning Via Instance Discrimination and Feature Decorrelation, ICLR, 2021.
>
> [3] Dual Mutual Information Constraints for Discriminative Clustering, AAAI, 2023.
>
> [4] Prototypical contrastive learning of unsupervised representations, ICLR, 2021.

---

> ### Author Response · Authors · 2024-11-21
> **Response to Reviewer EuZq (2/3)**
>
> ## Response to Reviewer EuZq (2/3)
> >### Weakness 2 and Question 2: Enhance the discussion on method interpretability by providing more empirical analysis regarding its impacts.
>
> **A**: Thank you for your valuable suggestion. We agree that besides the theoretical analysis, empirical analysis is also crucial for understanding the actual impact of ConMix.
>
> Following your advice, we conducted experiments on CIFAR-10 with imbalance ratios (IR) of 1, 2, 5, 10, 20, 50, and 100. And we provide the empirical analysis from the perspective of the compactness of the learned representations for different classes. Specifically, we train the baseline SimCLR and ConMix on CIFAR-10 under different imbalance ratios and calculate the class-wise similarity for each class. Class-wise similarity denotes the average similarity of samples within the same class and can **indicate the compactness of representations of different classes** in the feature space. Thus, to facilitate the description, we will refer to this metric as "**compactness**" below. We categorize the 10 classes of CIFAR-10 into Many, Medium, and Few categories based on the number of samples, following a 3:4:3 ratio. We first calculate the compactness for each class, then compute the average compactness for the three categories.
>
> Moreover, we propose a new metric, denoted as F/My, which represents the ratio of the Few category compactness to the Many's, to measure the balance between head classes (Many) and tail classes (Few). The larger F/My is, the greater the discrepancy in compactness between head classes and tail classes is, indicating a more severe impact of the long-tailed distribution. The results are shown in the below table.
>
> |                    | Many |Medium| Few  | F/My |
> |--------------------|------|------|------|------|
> | SimCLR (IR=1)      | 0.08 | 0.08 | 0.11 | 1.39 |
> | ConMix (IR=1)      | 0.16 | 0.16 | 0.20 | 1.25 |
> | SimCLR (IR=2)      | 0.05 | 0.08 | 0.11 | 2.25 |
> | ConMix (IR=2)      | 0.13 | 0.16 | 0.26 | 1.98 |
> | SimCLR (IR=5)      | 0.03 | 0.08 | 0.16 | 4.74 |
> | ConMix (IR=5)      | 0.10 | 0.17 | 0.35 | 3.36 |
> | SimCLR (IR=10)     | 0.03 | 0.08 | 0.22 | 7.74 |
> | ConMix (IR=10)     | 0.08 | 0.18 | 0.43 | 4.85 |
> | SimCLR (IR=20)     | 0.02 | 0.09 | 0.26 | 10.36|
> | ConMix (IR=20)     | 0.08 | 0.20 | 0.46 | 5.69 |
> | SimCLR (IR=50)     | 0.02 | 0.12 | 0.27 | 13.11|
> | ConMix (IR=50)     | 0.07 | 0.24 | 0.36 | 5.08 |
> | SimCLR (IR=100)    | 0.02 | 0.13 | 0.25 | 12.55|
> | ConMix (IR=100)    | 0.07 | 0.26 | 0.30 | 4.22 |
>
> From the table, we can derive the following empirical analysis:
>
> (1) The head classes (Many) typically have smaller compactness values, while tail classes (Few) have larger compactness values. This suggests that due to long-tailed effect, head classes occupy more of the feature space than tail classes.
>
> (2) When the distribution is long-tailed, the compactness for Few category is relatively higher, while the compactness for Many category is relatively lower, leading to a larger F/My ratio. This is a negative effect of the long-tailed distribution. However, ConMix can reduce the F/My ratio compared to SimCLR under different imbalance ratios, indicating its ability to mitigate the impact of the long-tailed distributions.
>
> (3) Regardless of whether the classes are Few, Medium, or Many, ConMix improves compactness across different imbalance ratios. This indicates that samples within the same class become more compact, which is beneficial for clustering.
>
> (4) When the imbalance ratio is 1, the dataset is balanced and differences in compactness across different classes are due to the varying difficulty of learning each class.
>
> The above analysis empirically demonstrates that ConMix benefits long-tailed clustering. We have included the above experimental results in Appendix F of the revised version. Thank you for your suggestion!

---

> ### Author Response · Authors · 2024-11-21
> **Response to Reviewer EuZq (3/3)**
>
> ## Response to Reviewer EuZq (3/3)
> >### Weakness 3 and Question 3: Provide detailed descriptions of the experimental setup and hyperparameter selections to improve transparency and reproducibility of the research.
>
> **A**：We agree that the transparency and reproducibility of the research are very important and we have described the experimental settings in Section 5.1. Based on your suggestion, we provide more detailed information on hyper-parameter choice and training specifics here.
>
> All experiments, unless otherwise specified, are conducted using ResNet18 with a batch size of 512 for 1000 epochs. As described in Section 5.1.2, we set the first convolution layer with kernel size 3×3 and stride 1, and remove the first max-pooling layer for all experiments on CIFAR-10 and CIFAR-20 due to their small image sizes.  For ConMix, we adopt the stochastic gradient descent (SGD) optimizer, whose learning rate is 0.5, weight decay is 0.0001 and momentum is 0.9. We adopt the cosine decay learning rate schedule to update the learning rate by step, with 10 epochs for learning rate warmup. During the 1000-epoch training, for the first 200 epochs, we train only with SDCLR to learn meaningful raw representations for interpolation. Then we only use ConMix to train the model. The temperature $\tau$ in NT-Xent is 0.2. We adopt the data augmentation methods described in [1]. Unless otherwise specified, synthesized representation number $M$ = 100 is used in the experiments.
>
> The above descriptions have been added to the revised paper in Section 5.1 and Appendix A.
>
> **Moreover, the code of our method has been submitted to the Supplementary Material, where these settings are already configured.** We will also open-source the code after the review, ensuring transparency and reproducibility.
>
> [1] A Simple Framework for Contrastive Learning of Visual Representation, ICML, 2020.
>
> Above is our response. Your suggestions and opinions are very important, and we are grateful for the time and effort you have devoted to improving this work. Thank you very much!

---

> ### Author Response · Authors · 2024-11-25
>
> Dear reviewer EuZq,
>
> Thanks again for your time and efforts in reviewing this paper and the valuable comments on improving its quality. As the reviewer-author discussion deadline approaches, we hope to hear your feedback about our response. If you have further concerns, we are happy to provide more explanations. Thank you very much!
>
> Regards from the authors.

---

> ### Author Response · Authors · 2024-11-26
> **Looking forward to your valuable reply**
>
> Dear Reviewer EuZq,
>
> We are very grateful for your valuable and constructive review comments. Given that the deadline for revising the paper is approaching, with less than two days remaining, we hope to learn whether our previous responses have met your satisfaction and if you have any additional questions or suggestions. If you have any new questions or suggestions, we will certainly provide a prompt response and make the corresponding revisions to the paper to improve it. We look forward to your reply with great anticipation. We would be very grateful for your reply amidst your busy schedule. Thank you for the time and effort you have invested in our work.
>
> Regards from the authors.

---

> ### Author Response · Authors · 2024-11-28
> **The deadline for revising the paper is approaching**
>
> Dear Reviewer EuZq,
>
> Thank you for the time and effort you have put into this work.
> The deadline for revising the paper is less than 12 hours away.
> Could you please take a few minutes to read our response?
> We look forward to knowing whether our response has addressed your concerns.
> If you have further concerns, we are happy to provide more explanations. Thank you very much!
>
> Regards from the authors.

---

> ### Author Response · Authors · 2024-12-02
>
> Dear Reviewer **EuZq**,
>
> As the Reviewer-Author discussion phase is drawing to a close, we kindly ask you to review our revisions and responses once more and reconsider your rating. All the other reviewers' concerns have been resolved, and they all gave this paper a positive score.
> We eagerly anticipate your feedback. Thank you.
>
> Best regards,
>
> The Authors

---

### Official Review · Reviewer_z1DH · 2024-11-05

**Soundness:** 3
**Presentation:** 2
**Contribution:** 3
**Rating:** 6
**Confidence:** 3

**Summary:**

This paper proposes a new method called ConMix for dealing with the long-tailed problem of deep clustering. A major challenge in long-tailed deep clustering is how to deal with class imbalance in a dataset without label information. ConMix solves this problem through an innovative approach to mixed representations in contrastive learning to enhance deep clustering performance in the case of long-tailed distributions.

**Strengths:**

1.	The author has conducted comprehensive experiments, compared with multiple clustering algorithms, and prove the effectiveness of ConMix under long-tailed distribution.
2.	Reasonable theoretical analysis is given to verify that ConMix can implicitly achieve the loss-balance.
3.	Contributions of different elements in ConMix are studied through extensive experiments.

**Weaknesses:**

The representation synthesis part is supposed to be represented more intuitively, which may be a little confusing at first reading.

**Questions:**

1. The result of pairwise ConMix shown in Table3 on CIFAR-10 is better than the result of ConMix with M=500 presented in Figure 2.Is this reasonable? In my understanding, the former is equivalent to ConMix with a larger M on CIFAR-10.
2. Have you conducted additional experiments on balanced models of other methods as ConMix-B to support the opinion about robustness?
Question3. Are there other clustering methods being studied besides k-means?

---

> ### Author Response · Authors · 2024-11-21
> **Response to Reviewer z1DH (1/3)**
>
> ## Response to Reviewer z1DH (1/3)
> Thank you for your constructive comments. We have carefully considered your review and provided the response as below.
>
> > ### Weakness 1: The representation synthesis part is supposed to be represented more intuitively, which may be a little confusing at first reading.
>
> **A**: Sorry for confusing you at first reading. In simple terms, we assign a tag within $[1, M]$ to each input within every batch, where $M$ is the number of synthesized representations. Then, representations from the same network branch with the same tag are averaged to form new representations in the manner of Eq. (3). The generation of tags follows a uniform distribution with equal probabilities $\frac{1}{M}$.
>
> We reviewed the original paper and made the following revisions to alleviate the readers' confusion.
>
> (1) We modified Line 208-209 from "In SimCLR framework...augmented twice" to:
>
> "In SimCLR framework, each input $x_i$ is data-augmented twice, and the two augmented versions are fed into two different network branches in SimCLR. Given $N$ inputs, the network will output $2N$ representations {$v_1, v_2, ...v_{2N}$}."
>
> (2) We modified Line 216-218 from "We randomly select...with equal contributions" to:
>
> "In each batch, we randomly assign tags within $[1, M]$ to original representations from the same network branch {$v_1, v_2, ...v_N$}. The generation of tags follows a uniform distribution with equal probabilities $\frac{1}{M}$ and original representations with the same tag are used to synthesize one particular representation in the manner of Eq. (3)."
>
> We hope these revisions can improve the readability of the paper. Thank you.
>
> >### Question 1: The result of pairwise ConMix shown in Table3 on CIFAR-10 is better than the result of ConMix with M=500 presented in Figure 2. Is this reasonable? In my understanding, the former is equivalent to ConMix with a larger M on CIFAR-10.
>
> **A**: The mentioned results in Table 3 and Figure 2 are two different experiments with different settings. Specifically, pairwise ConMix in Table 3 is used to validate the effectiveness of the multi-sampling strategy compared with pairwise sampling. While the experiment with $M$=500 in Figure 2 is to evaluate the impact of different $M$.
>
> The significant performance drops in Figure 2 when $M$=500 might confuse you. But the shown results are actually reasonable. The reason lies in the batch size being 512. **So when synthesizing $M$=500 representations within each batch, the multi-sampling strategy in ConMix almost fails to work effectively.** But pairwise ConMix synthesizes 256 (half of 512) new representations and still has good performance.
>
> We have emphasized in Appendix B that the batch size is 512, to facilitate better understanding. We have made the following revision:
>
> "Due to the batch size of 512, when $M$ is set to 500, the multi-sample approach in ConMix diminishes considerably, leading to a drop in experimental performance."

---

> > ### Author Response · Authors · 2024-11-21
> > **Response to Reviewer z1DH (2/3)**
> >
> > ## Response to Reviewer z1DH (2/3)
> >
> > >### Question 2: Have you conducted additional experiments on balanced models of other methods as ConMix-B to support the opinion about robustness?
> >
> > **A**: Thank you for this valuable suggestion. Following your suggestion, we further trained and tested the performance of different methods on balanced datasets. The dataset configurations and experimental setups refer to [1]. Specifically, we trained different methods using ResNet-18 for 1000 epochs on CIFAR-10 and CIFAR-20. Since ConMix does not leverage some advanced techniques suitable for balanced clustering, we propose an updated version of ConMix called "ConMix+Propos" that embeds ConMix into Propos [1]. We first train the model with the loss of ConMix for 500 epochs, then train it with Propos for another 500 epochs. The total number of training epochs for this updated version is the same as other methods.
> >
> > We have reported the results of the balanced dataset in the following tables, and also provided the clustering performance on the datasets with an imbalance ratio of 10 in the **parentheses** ( $\cdot$ ), for better comparisons.
> >
> > The experimental results are as follows.
> >
> > CIFAR-10:
> >
> > |         | ACC         | NMI         | ARI         |
> > |---------|-------------|-------------|-------------|
> > | SimCLR  | 72.8 (39.4) | 63.9 (38.4) | 56.7 (23.7) |
> > | SDCLR   | 71.4 (38.9) | 62.4 (42.5) | 54.8 (26.5) |
> > | CC      | 79.0 (40.6) | 70.5 (43.9) | 63.7 (18.8) |
> > | IDFD    | 81.5 (47.5) | 71.1 (48.4) | 66.3 (33.1) |
> > | Propos  | 91.6 (46.1) | 85.1 (52.5) | **83.5** (34.2) |
> > | ConMix  | 80.9 (53.3) | 70.7 (57.1) | 65.6 (40.8) |
> > | ConMix+Propos | **92.0** (**53.8**) | **85.3** (**58.8**) | 83.3 (**42.8**) |
> >
> >
> > CIFAR-20:
> >
> > |         | ACC         | NMI         | ARI         |
> > |---------|-------------|-------------|-------------|
> > | SimCLR  | 45.4 (34.4) | 43.8 (36.9) | 28.8 (19.8) |
> > | SDCLR   | 44.6 (37.8) | 43.3 (39.6) | 27.6 (22.9) |
> > | CC      | 42.9 (19.9) | 43.1 (21.9) | 26.6 ( 1.1) |
> > | IDFD    | 42.5 (28.7) | 42.6 (28.6) | 26.4 (15.1) |
> > | Propos  | 57.8 (36.8) | 58.2 (40.1) | 42.3 (22.5) |
> > | ConMix  | 46.0 (41.7) | 45.5 (43.6) | 29.8 (27.0) |
> > | ConMix+Propos | **59.2** (**43.8**) | **58.4** (**47.4**) | **42.5** (**30.3**) |
> >
> > Compared to the baseline methods (SimCLR, SDCLR), ConMix **demonstrates significant performance improvements on both the balanced datasets and the long-tailed datasets.** Compared to recent deep clustering methods (CC, IDFD, Propos), ConMix shows improvements on long-tailed datasets but may not perform as effective as Propos on balanced datasets. Note that, ConMix still outperforms CC and IDFD on most cases even on the balanced datasets.
> >
> > However, the updated version "ConMix+Propos" performs the best on both balanced datasets and long-tailed datasets, showing that adding some recent deep clustering techniques on "ConMix" will further improve its performance and robustness.
> >
> > Moreover, we also notice that existing deep clustering algorithms perform well on balanced datasets **but suffer severe performance degradation on long-tailed datasets.** We believe this is due to these methods making assumptions that are aligned with balanced datasets. While they can achieve good laboratory performance, they are less suitable for realistic long-tailed distributions. This further underscores the importance of research on long-tailed deep clustering.
> >
> > The above results and analyses have been added in Appendix C of the updated paper to demonstrate robustness of our work. Thank you.
> >
> > [1] Learning Representation for Clustering Via Prototype Scattering and Positive Sampling, TPAMI, 2023.

---

> > > ### Author Response · Authors · 2024-11-21
> > > **Response to Reviewer z1DH (3/3)**
> > >
> > > ## Response to Reviewer z1DH (3/3)
> > > > ### Question 3: Are there other clustering methods being studied besides k-means?
> > >
> > > **A**: Our primary consideration for using K-means as the clustering method is that most of our compared methods utilize K-means (except for Contrastive Clustering which directly outputs the clustering assignments). So the use of K-means allows us to achieve a fair comparison.
> > >
> > > However, we did investigate the impact of different clustering methods on the performance of ConMix. For example, we tried the Gaussian Mixture Model (GMM) [1] on the learned embedding of ConMix to obtain cluster assignments. We use two methods to initialize GMM: one is to initialize using K-means, and the other is random initialization. We denote these two cases as GMM-k and GMM-r. Besides, we tested the performance of agglomerative clustering [2] on ConMix (in the table, it is abbreviated as AC). The detailed results are as follows.
> > >
> > > CIFAR-10 with an imbalance ratio of 10:
> > >
> > > | Metric  | ACC  | CAA  | NMI  | ARI  |
> > > |---------|------|------|------|------|
> > > | K-means | 53.3 | 58.2 | 57.1 | 40.8 |
> > > | GMM-k   | **63.6** | 48.1 | **59.1** | **53.1** |
> > > | GMM-r   | 50.3 | 49.0 | 58.7 | 44.7 |
> > > | AC      | 59.8 | **62.2** | 58.8 | 46.5 |
> > >
> > > CIFAR-20 with an imbalance ratio of 10:
> > >
> > > | Metric  | ACC  | CAA  | NMI  | ARI  |
> > > |---------|------|------|------|------|
> > > | K-means | **41.7** | **39.3** | **43.6** | **27.0** |
> > > | GMM-k   | 37.2 | 35.4 | 42.7 | 23.7 |
> > > | GMM-r   | 31.3 | 22.2 | 34.9 | 18.1 |
> > > | AC      | 39.0 | 37.0 | 42.0 | 20.0 |
> > >
> > > STL-10 with an imbalance ratio of 10:
> > >
> > > | Metric  | ACC  | CAA  | NMI  | ARI  |
> > > |---------|------|------|------|------|
> > > | K-means | 47.4 | 48.4 | 48.2 | 33.9 |
> > > | GMM-k   | **50.6** | 40.7 | 47.3 | **34.1** |
> > > | GMM-r   | 45.9 | 46.0 | 46.2 | 31.5 |
> > > | AC      | 47.4 | **51.4** | **49.7** | 33.5 |
> > >
> > >
> > > Compared with these methods, we can find that different clustering methods can all lead to good performance, validating the effectiveness of our method ConMix. However, for different datasets, the methods achieving the best performance may vary. For example, on CIFAR-10, GMM-k achieved the highest ACC, NMI and ARI, while AC achieved the highest CAA. On CIFAR-20, K-means performed the best. On STL-10, GMM-k obtained the best results in terms of ACC and ARI, while AC achieved the highest CAA and NMI.
> > >
> > > Considering that the compared methods use K-means mostly, to ensure a fair comparison, we report the results of using K-means in the paper. Thank you for your valuable question and we have added the above results in Appendix D.
> > >
> > > [1] Gaussian mixture models, Encyclopedia of biometrics, 2009.
> > >
> > > [2] Hierarchical agglomerative clustering procedure, Pattern Recognition, 1979.
> > >
> > > Above is our response. We are grateful for your thorough review and valuable suggestions. Thank you very much!

---

> ### Author Response · Authors · 2024-11-25
>
> Dear reviewer z1DH,
>
> Thanks again for your time and efforts in reviewing this paper and the valuable comments on improving its quality. As the reviewer-author discussion deadline approaches, we hope to hear your feedback about our response. If you have further concerns, we are happy to provide more explanations. Thank you very much!
>
> Regards from the authors.

---

> ### Author Response · Authors · 2024-11-26
> **Looking forward to your valuable reply**
>
> Dear Reviewer z1DH,
>
> We are very grateful for your valuable and constructive review comments. Given that the deadline for revising the paper is approaching, with less than two days remaining, we hope to learn whether our previous responses have met your satisfaction and if you have any additional questions or suggestions. If you have any new questions or suggestions, we will certainly provide a prompt response and make the corresponding revisions to the paper to improve it. We look forward to your reply with great anticipation. We would be very grateful for your reply amidst your busy schedule. Thank you for the time and effort you have invested in our work.
>
> Regards from the authors.

---

### Author Response · Authors · 2024-12-04
**Summary of Our Responses**

We thank all the reviewers (**z1DH**, **EuZq**, **RUUY**) for their efforts in improving this work.

Overall, the reviewers acknowledged the following strengths of our work:

(1) A notable contribution for a method specially designed for long-tailed deep clustering,
a field that has received little attention in previous research. (**EuZq**, **RUUY**)

(2) The innovation of extending mixup to unsupervised learning and long-tailed learning. (**EuZq**, **RUUY**)

(3) Reasonable theoretical analysis which adds depth to the paper. (**z1DH**, **EuZq**, **RUUY**)

(4) Comprehensive experiments which prove the effectiveness of ConMix. (**z1DH**, **EuZq**, **RUUY**)

Meanwhile, following the reviewers' suggestions, our main revisions are as follows:

(1) We have clarified content in the paper that might cause misunderstandings and revised the manuscript accordingly
to enhance its readability. (**z1DH**, **RUUY**)

(2) We have added results on balanced datasets to prove the robustness of ConMix. (**z1DH**, **RUUY**)

(3) We investigated the performance of clustering techniques besides K-means. (**z1DH**)

(4) We have added results on ImageNet to further demonstrate the effectiveness and generalization capability
of ConMix. (**EuZq**)

(5) We conducted experiments on datasets with varying imbalance ratios and provide empirical analysis for
understanding the actual impact of ConMix on the learning process. (**EuZq**)

We believe our extensive experiments and detailed responses have addressed the reviewers' concerns.

Above is the summary of our responses. Thank you for your reading!

---

### Meta-Review · Area_Chair_dqVn · 2024-12-19

**Metareview:**

This paper proposes a deep clustering method for imbalanced data. Such a long-tail problem is less explored in clustering research but is common in real-world applications. The proposed method is simple yet effective, with detailed theoretical analyses provided by the authors. As all three reviewers gave positive scores, I decided to accept this paper. The authors should further improve the clarity of the manuscript in its camera-ready version to help readers interpret and reproduce the proposed method.

**Additional Comments On Reviewer Discussion:**

In the reviewer discussion session, the reviewers were satisfied with the authors' response and raised their scores accordingly. One reviewer did not respond, but after reading the authors' rebuttal, I feel the concerns have been addressed. In the rebuttal period, the authors further demonstrated that the proposed method could be incorporated into existing deep clustering methods, thereby strengthening the contributions of this work.

---

### Decision · Program_Chairs · 2025-01-22

Accept (Poster)